# Ship Emissions and the use of current air cleaning technology: Contributions to air pollution and acidification in the Baltic Sea.

Björn Claremar, Karin Haglund, Anna Rutgersson

Department of Earth Sciences, Uppsala University, Uppsala, 75236, Sweden

*Correspondence to*: Anna Rutgersson (anna.rutgersson@met.uu.se)

**Abstract**. The shipping sector is a significant contributor to emissions of air pollutants in marine and coastal regions. In order to achieve sustainable shipping, primarily through new regulations and techniques, greater knowledge of dispersion and deposition of air pollutants is required. Regional model calculations of the dispersion and concentration of sulphur, nitrogen and particulate matter, as well as deposition of oxidized sulphur and nitrogen from the international maritime sector in the Baltic Sea and the North Sea have been made for the years 2011 to 2013. Contribution from shipping is highest along shipping lanes and near large ports for concentration and dry deposition. Sulphur is the most important pollutant coupled to shipping. The contribution of both $SO_2$ concentration and dry deposition of sulphur represented up to 80% of the total in some regions. WHO guidelines for annual concentrations were not trespassed for any analysed pollutant, other than $PM_{2.5}$ in the Netherlands and Belgium, and central Poland. But due to the resolution of the numerical model, 50 x 50 km$^2$, there may be higher concentrations locally close to intense shipping lanes. Wet deposition is more spread and less sensitive to model resolution. The contribution of wet deposition of sulphur and nitrogen from shipping was up to 30% of the total wet deposition. Comparison of simulated to measured concentration at two coastal stations close to shipping lanes showed some underestimations and missed maximums, probably due to resolution of the model and underestimated ship emissions.

Changed regulation for maximum sulphur content in maritime fuel, in 2015 from 1% to 0.1%, decreases the atmospheric sulphur concentration and deposition significantly. But due to costs related to refining, the cleaning of exhausts through scrubbers has become a possible economic solution. Open-loop scrubbers meet the air quality criteria but their consequences for the marine environment are largely unknown. The resulting potential of future acidification in the Baltic Sea, both from atmospheric deposition and from scrubber water along the shipping lanes, based on different assumptions about sulphur content in fuel, scrubber usage and increased shipping density has been assessed. The increase in deposition for different shipping and scrubber scenarios differs for the basins in the Baltic Sea, with highest potential of acidification in the southern basins with high traffic. The proportion of ocean acidifying sulphur from ships increases when taking scrubber water into account and the major reason to increasing acidifying nitrogen from ships are due to increasing ship traffic. Also with the implementation of emissions control for nitrogen, the effect of scrubber on acidification is evident. This study also generates a database of shipping and scrubber scenarios for atmospheric deposition and scrubber exhaust from the period 2011 to 2050.

*Keywords:* Sulphur dioxide, nitrogen dioxide, nitrogen oxides, particulate matter, EMEP model, deposition, shipping, air pollutants, scrubber, Baltic Sea region.

## 1 Introduction

Emissions of air pollutants is a large problem, air pollutants have harmful effects on human health, the environment and buildings. They also influence climate and water quality (Seinfeld and Pandis, 2006; Monks et al., 2009; Fuglestvedt and Berntsen, 2009). There has been a significant decrease in land based emissions over land areas in Europe since the risks associated with high levels of air pollutants were brought into light two decades ago. During the same time, however, emissions from shipping in the Baltic Sea and the North Sea have increased, with the exception of a very recent decrease in sulphur emissions and successive emissions of particulate matter due to regulations (Gauss et al., 2013; Jonson et al., 2015). Shipping is the most cost-effective option for global transport of goods, and over 90% of the world trade is carried by sea (IMO, 2016). The Baltic Sea area is one of the busiest shipping areas in the world and it is of great importance for the development and economy of the surrounding countries. The intensity of shipping in the Baltic Sea has increased during the last decade and it is expected to increase further in the coming years.

Shipping primarily generates emissions of nitrogen oxides ($NO_x$), sulphur dioxide ($SO_2$), carbon monoxide (CO), carbon dioxide ($CO_2$), volatile organic compounds (VOC) and particulate matter (PM) (Corbett and Fischbeck, 1997; Eyring et al., 2010; Matthias et al., 2010). Maritime contribution of sulphur dioxide into the atmosphere is mainly caused by the high sulphur content in the fossil fuels used by the sector (Eyring et al., 2005). Nitrogen oxides include nitric oxide (NO) and nitrogen dioxide ($NO_2$), which are emitted from engines that operate under high temperature and pressure (Eyring et al., 2010). Particulate matter from shipping consists of a complex mixture of sulphate ($SO_4$), soot, metals and other organic and inorganic fragments (Winnes et al., 2014), the prime component being sulphate, which is formed by oxidation of $SO_2$ (Eyring et al., 2010). The quantity and size of particulate matter emitted from shipping depends mainly on the type of fuel and its sulphur content, as well as the ship's engine (Fridell et al., 2008; Aardenne et al., 2013). Particulate matter is divided into $PM_{10}$ and $PM_{2.5}$ in terms of its aerodynamic diameter where $PM_{10}$ has an aerodynamic diameter less than 10 micrometres, while $PM_{2.5}$ has a diameter less than 2.5 micrometres. $SO_2$ is also chemically transformed into sulphuric acid in the presence of liquid water or water vapour and can cause acid rain which contributes to the acidification of the oceans, lakes and soil. Sulphur and nitrogen from oxides are called oxidized sulphur, OXS, and oxidized nitrogen, OXN, in deposition and both act as acidifying compounds.

Effects of air pollution vary in both space and time; they may be short-lived and local or more prolonged and global (Seinfeld and Pandis, 2006). Exposure to particulate matter encompasses a variety of risks to human health, primarily on the respiratory organs and the cardiovascular system (World Health Organization, 2006). Corbett et al. (2007) estimated that shipping-related emissions of particulate matter contribute to approximately 60,000 deaths annually on a global scale, with impacts concentrated to coastal areas along the major trade routes. Further, particulate matter may be absorbing or reflecting which has an impact on the Earth's radiation balance. The net effect of emissions from the maritime sector on the radiation balance is negative, resulting in a cooling effect (Eyring et al., 2005; Fuglestvedt and Berntsen, 2009). Jonson et al. (2015), hereafter abbreviated J15, estimated that current emissions from shipping in the Baltic Sea region cause a life loss per person by 0.1–0.2 years, in areas close to the main shipping lanes. Exposure to high levels of sulphur oxides cause health issues such as irritation to respiratory system, lungs and eyes (World Health Organization, 2006). High levels of nitrogen in the atmosphere also have negative impacts on human health, cause corrosion of materials and are included in the process of degrading of methane (Fuglestvedt and Berntsen, 2009; Eyring et al., 2010). Deposition of nitrate contributes to

both eutrophication and acidification of water and soil. A pH reduction in the ocean causes worsening conditions to a lot of marine ecosystems (Andersson et al., 2008). Hunter et al. (2011) modelled acidification from strong acids in the North Sea, Baltic Sea and South China Sea. On an annual scale the trends were on the order of $10^{-4}$ pH units per year, comparable to the global assessment by Doney et al. (2007). Hassellöv et al. (2013) modelled pH decrease from shipping, worldwide, on the seasonal scale. In areas with heavy ship traffic and seasonal stratification of the surface water gave larger pH decrease, comparable to the effects from $CO_2$ uptake. Thus locally the annual decrease is of the order of 0.002 pH units (Orr, 2011, Rhein et al. 2013). On the annual scale the results from Hassellöv et al. (2013) is comparable with Hunter et al. (2011) and Doney et al. (2007). Acidification is a major challenge in the Baltic Sea region today where the critical load is exceeded in big parts of the area (Gauss et al., 2013). Due to its brackish water the Baltic Sea has a rather lower buffer capacity, and is thus more sensitive to acidification (Andersen et al., 2010).

The maritime sector was, at least earlier, one of the least controlled sources of anthropogenic emissions. It is a global cross-border sector with conditions making legislation challenging (Aardenne et al., 2013). The International Maritime Organization (IMO) is the agency within the United Nations (UN) responsible for maritime security and safety together with prevention of pollutants by ships (International Maritime Organization, 2015). IMO has formulated The International Convention on the Prevention of Pollution by Ships (MARPOL) which has been ratified globally (CleanShip, 2013). MARPOL and its Annex VI regulate emissions from ships.

The regulations include the Sulphur Emission Control Area (SECA) which consists of the Baltic Sea, North Sea, English Channel and North America's coastal areas (International Maritime Organization, 2015, J15). Residual oil or High Sulphur Fuel Oil (HSFO) can have up to 3.5% sulphur content, but a global mean estimate is 2.4% (IMO, 2011) to 2.7% outside SECA (ENTEC, 2005). In the SECAs a number of reductions have been made. In May 2006 the sulphur content in maritime fuel was restricted to 1.5% (percentage by mass) by refining to marine gas oil (MGO). In 2010 it was reduced to 1.0% and according to J15, this reduction of sulphur had a positive effect on air quality and the deposition of sulphur. A further reduction of the permitted level of sulphur to 0.1% was made in January 2015 (Aardenne et al., 2013). From 1 January 2020 the upper sulphur content should be reduced to 0.5% globally (outside the SECAs) even if 2025 is a more probable outside EU (Jonson et al. 2015). The reduction in SECA has led to extensive investment in scrubbers, or Exhaust Gas Cleaning Systems (EGCS), since the refined oil increases in price. Scrubbers use seawater to remove the sulphur oxides generated from high-sulphur fuels. An expected effect of open-loop scrubbers is that acidification is concentrated along the shipping lanes as the scrubber exhaust is released into the water. With scrubbers the ships can still use HSFO and it seems that the 0.5% limit can be walked around (S&P Global Platts, 2016).

There is currently no international regulation of direct particulate emissions from shipping. But with less sulphur, also particle emissions will decrease, but since there are other sources as well, the decrease is less. The regulations of nitrogen emissions in MARPOL (TIER) are defined as a function of year of installation and ship speed (IMO, 2007). The TIER I standard was implemented in 2000 and was 10% stricter than for ships built before 2000. The introduction of TIER II in 2011 was up to 15% stricter then the former. As regulations only include newly produced ships, the effects of regulations of nitrogen emissions from shipping have so far been small. There are also Nitrogen Emission Control Areas (NECA), e.g. in the Caribbean Sea and along the North American coast, but so far, not in the Baltic Sea and North Sea. However, at the end of 2016 it was decided that in 2021 a NECA will be introduced for the Baltic Sea and North Sea. There are indications that with this introduction, emissions of $NO_x$ will decrease

by 80% relative to the TIER I level (J15). In J15 it was assumed that half of the fleet followed the TIER III commitment by 2030, and all in 30 years (2050). In addition to HSFO and low sulphur oil there are also other possible fuels, like liquefied natural gas (LNG) and methanol.

J15 did a thorough study of air pollutants from shipping in the Baltic Sea and the North Sea. They used the EMEP model (Simpson et al., 2012) and estimated the effects of present and future emissions of $NO_2$, $SO_2$ and particles as PM10 and PM2.5, among other compounds. As ship emission data they used the ship traffic Emission Assessment Model (STEAM) (Jalkanen et al., 2009, 2012). This model is based on movements of individual ships with high temporal resolution (Automatic Identification System, AIS) together with engine times and corresponding emission factors. The concentrations and depositions were analysed for the year 2010. Concentrations were focused along the shipping routes but there was a significant spread for depositions. The effect of the reduced sulphur content in the fuel made in 2010 and the effect of future scenarios with reduced sulphur content (2015 value of 0.1%) and on regulation of NECA was also investigated.

The present study is to some extent similar to J15, we study deposition and near surface concentrations originating from shipping, introducing also the estimation of the future effects from scrubbers in the Baltic Sea. We use the same chemical transport model as J15, the EMEP model, although at a lower resolution, 50 x 50 km$^2$, compared to 14 x 14 km$^2$. There are some differences compared to J15, we use default ship emission data for the EMEP model whereas J15 use AIS based ship emissions. We analyse the present concentration due to ship emissions and its deposition but focus on the sulphur exhausts in a future scenario relying on scrubber technique (defining suggested scenarios for use of scrubbers). The model is run for five years (2009 to 2013) and by using scenarios for future shipping and cleaning technologies, estimates of deposition (from air and scrubber) into the Baltic Sea until 2050 are derived. The use and averaging of 3 years (2009–2011) for the present deposition fields reduces the variability from meteorology for the future scenarios. The scenarios are limited to "worst case scenarios" regarding the use of scrubbers, but the results will be discussed in relation to other possibilities. In the analysis of the period around 2010 we examine the impact of having coarse resolution on concentration and deposition as well as using non-AIS databases for ship emissions, when comparing our results to J15. Concentrations may, for instance be very dependent on resolution whereas deposition may be less sensitive.

**2 Data**

**2.1 EMEP Model System**

The unified European Monitoring and Evaluation Programme (EMEP) is policy driven under the Convention on Long-range Transboundary Air Pollution (CLRTAP) for international co-operation to solve transboundary air pollution problems. The programme is divided into five centres working with emission inventories, measurements, and chemical and dispersion modelling (http://www.emep.int/) The EMEP model is a chemical atmospheric transport model (http://emep.int/mscw/index_mscw.html). The model is Eulerian and traditionally consists of a three-dimensional grid that covers Europe. The standard horizontal resolution is approximately 50 km $\times$ 50 km at 60°N and has 20 layers in the vertical direction up to 100 hPa. Land use is separated into 16 classes. Emissions included in the EMEP model are sulphur dioxide, nitrogen oxides, ammonia ($NH_3$), non-methane volatile organic compounds (NMVOC), carbon monoxide and particulate matter. The model´s lateral boundary concentrations

consist of a merging of observed data and results from global models. A more comprehensive description of the EMEP model can be found in Simpson et al. (2012).

The EMEP model is considered to be a robust model for dispersion modelling in the atmosphere (Simpson et al., 2012; Gauss et al., 2015). In Gauss et al., 2015 a comparison of model results from the EMEP model version rv.4.7 and observations of annual averages of concentrations at individual stations for 2013 were made. On average, sulphur dioxide concentration was underestimated by 11%, nitrogen dioxide was underestimated by 7%, $PM_{10}$ was underestimated by 28% and $PM_{2.5}$ by 19%. Validation of wet and dry deposition of oxidized nitrogen and sulphur based on approximately 30 test sites in 2013 shows, despite the limitations of the model, a relatively good agreement with observed data considering a low bias and good correlation (Gauss et al., 2015).

**2.2 EMEP Model Data**

The meteorological input data used in the EMEP model are from the Integrated Forecast System (IFS) which is a global forecast model run by the European Centre for Medium-Range Weather Forecasts (ECMWF). Chemical data used in the EMEP model cover 56 persistent and 15 short-lived components, chemical reactions, phase changes and solubility in water. Emission inputs consist of gridded yearly national emission data (Vestreng, 2003; Simpson et al., 2012). The anthropogenic emissions are categorized in ten different groups called Selected Nomenclature for reporting of Air Pollutants (SNAP). All nations in the EMEP-area are responsible for reporting annual gridded emission data for each SNAP sector. National shipping is included in SNAP 8 (Other mobile sources and machinery) and is a part of the emission data that each nation should report. International shipping is also included in SNAP 8. International ship emission data used in the model were designed according to Table 1. In EMEP model rv.4.4 ship emissions and their distributions were mainly ENTEC/IIASA inventory. ENTEC UK Ltd. (now part of AMEC Environment Infrastructure, UK, www.amec-ukenvironment.com) compiled an emission inventory for 2000 (ENTEC, 2002) based on Lloyd's Register (1998). In addition, data on ship activity in ports were acquired using questionnaires (ENTEC, 2002). IIASA has updated these data in recent years with trends (Cofala et al., 2007).

In the development of the data set in EMEP model version rv.4.8 new aspects, as SECA, the economic situation and the using of different sizes of ships have been included. The emission data were designed for 2000 to 2011. In order to supplement emission data for the following years, extrapolation with Centre on Emission Inventories and Projections (CEIP) method were used. This interpolation was, however, shown to significantly underestimate the 2012 and 2013 emissions in the North Sea and in the Baltic Sea (Fagerli et al., 2015). Emissions from international shipping are assumed to be constant throughout the day and year in the model (Simpson et al., 2012). This was also shown by Jalkanen et al. (2014) being within 10% in the years 2006-2009.

Between model versions several changes that affect aerosol production/modelling have been implemented by the EMEP community; e.g. modification of the sea salt parametrisation, changes in the standard aerosol surface area and uptake rates, dust boundary conditions and an update of the split of particulate matter into elemental carbon, organic matter and the remainder. Furthermore, biogenic emissions of dimethyl sulphide (DMS) have been updated. Rather than being prescribed, DMS emissions are now calculated dynamically during the model calculation and vary with meteorological conditions.

Comparing the emissions in rv.4.4 and 4.8 shows very small differences, on the order of less than 1%. However, deposition and concentrations deviate significantly due to the new modifications of the chemistry and physics. Fig.

1 shows the relative change in 2011 (in %) going from rv.4.4 to 4.8 in deposition (from shipping) of oxidised

sulphur (OXS) and nitrogen (OXN), respectively. Overall, the depositions, has increased, but mostly over land,

and OXN in northern Atlantic and for OXS north of Great Britain. In Baltic Sea the increase is minor for OXS,

being less than 10%, but for OXN between 5 and 30%.

**Table 1**. International ship emission data for the different versions of the EMEP model used in this study

| EMEP model version | Simulated years | International ship emission data |
| --- | --- | --- |
| rv.4.4 | 2009-2011 | ENTEC international shipping data (Jonson et al., 2009; ENTEC, 2010) and trends after 2000 are from IIASA (Cofala et al., 2007) |
| rv.4.8 | 2011-2013 | Based on data developed by TNO in the EU Horizon 2020 project MACC III (Gauss et al., 2015) |

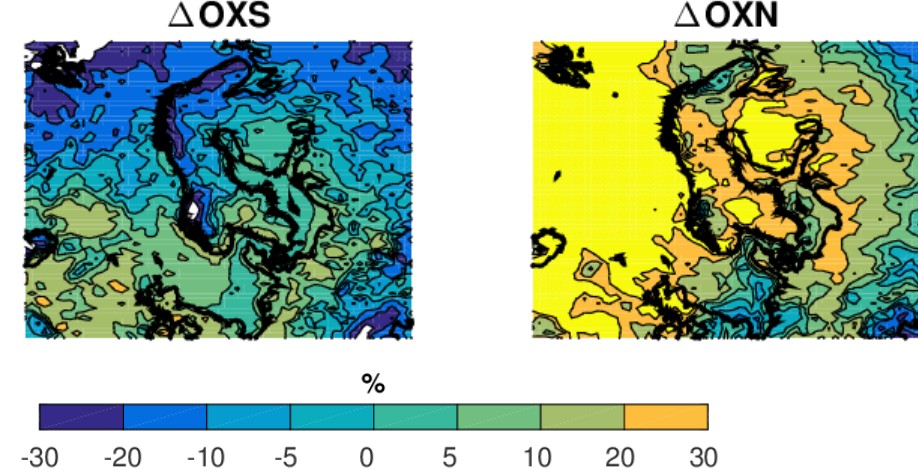

**Figure 1.** Difference (in %) of deposition due to shipping, from EMEP model rv.4.4 to rv.4.8 run for the year 2011 for a) oxidized sulphur and b) oxidized nitrogen.

**2.3 Other data sources**
Historical ship emission data were taken from the global gridded EDGAR 4.2 dataset (Olivier et al. 2011), with a
horizontal resolution of 0.1° and available from 1970 to 2008 (http://edgar.jrc. ec.europa.eu). For the 1900–1969
period, the EDGAR–HYDE 1.3 dataset has a resolution of 1° (Van Aardenne et al., 2001). Both these datasets
contain global anthropogenic emissions of NOx, SO2, NH3, and other species. The EDGAR-HYDE data are
derived from historical activity data from the 10-y-interval Hundred Year Database for Integrated Environmental
Assessments (HYDE), 1890–1990. These data are based on the data and methodology of EDGAR 2.0 (Olivier et
al., 1996). Linear interpolation of the emissions was used to fill the gaps between the 10-y intervals of EDGAR–
HYDE 1.3.
For background emissions of $SO_2$, NOx, and the deposition of OXS, OXN (i.e., from sources other than ship
traffic), output from the MATCH chemical transport model (Robertson et al. 1999) was used. We used a simulation
for the 1900–2050 period set up as described by Engardt and Langner (2013). Forcing was based on the RCP4.5
radiative scenario and accompanying anthropogenic emissions (Lamarque et al., 2010). Shipping emissions were
from Eyring et al. (2010) and the International Comprehensive Ocean-Atmosphere Data (ICOADS) spatial proxies
were used
**2.4 Measurements**
We validated the EMEP modelled data for 2013 using measured concentrations of nitrogen dioxide, sulphur
dioxide and particulate matter from stations Vavihill and Utö (Fig. 2). Data for Vavihill were downloaded from
the database of the Swedish Environmental Research Institute
(http://www.ivl.se/sidor/omraden/miljodata/luftkvalitet.html) and data for Utö were from the Finnish
Meteorological Institute's website (http://www.ilmanlaatu.fi/tarkistetut_tulokset/). The measuring station in
Vavihill is located in Svalöv municipality (N 56,142°; E 13,855°), 28 km from the port city of Helsingborg and 25
km from the coast of Øresund (Fig. 2). Within a radius of 10 km from the measuring station, no emission sources
that are assumed to have a significant impact on air quality are located. At the distance of 10 km from the measuring
station, there is a heavily trafficked road and within 50 km the larger cities of Lund and Malmö are located (Sjöberg
and Peterson, 2014; IVL Swedish Environmental Research Institute, 2015). The measuring station on Utö is
located in the central parts of the island (N 59,779°; E 21,394°), a few hundred m from the shore (Fig. 2). About
300 m away from the test site, there is a smaller shipping lane and a harbour for small boats. About 10 km west of
the measurement site, there is an international shipping lane that is heavily trafficked by larger vessels (Finnish
Meteorological Institute, 2015).

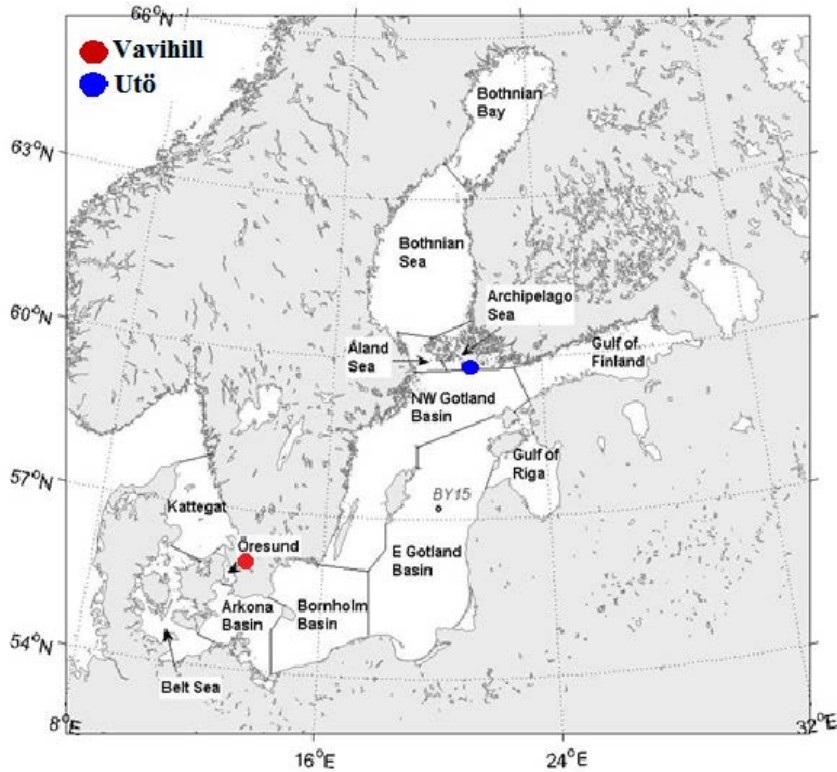

**Figure 2.** The division of the basins of the Baltic Sea-Skagerrak system, OR = Øresund, GO = Eastern Gotland
Basin, AL = Åland Sea, BE = Belt Sea, NW = North West Gotland Basin, AS = Archipelago Sea , AR = Arkona
Basin, GR = Gulf of Riga , BS = Bothnian Sea , KA = Kattegat, BH = Bornholm Basin, GF = Gulf of Finland, and
BB = Bothnian Bay.  Dots represent the measuring stations at Vavihill and Utö. Figure is redrawn from Omstedt
et al., (2015).
**3 Methods**
**3.1 EMEP Model Runs**
This investigation consists of two parts. A database with historical and future scenarios of emissions and
depositions of oxidized sulphur and nitrogen (the data base) was created. Model version rv.4.4 was used with
meteorology from the years 2009 to 2011 and emissions from 2011 (see sec 3.3). In the second part, the model
was validated to coastal measurements of concentrations for the year 2013 using the newer model version rv.4.8.
As seen in sec 2.2 rv.4.8 gives higher depositions of OXS and especially OXN. This is further discussed in sec 5.
Also the spatial pattern of concentration and deposition was analysed and compared to the results from J15. For
each studied year, two model runs in the EMEP model were made, a base run and a scenario run. In the base run,
all emission sources were included, and in the scenario run, the emissions from international shipping in the Baltic
Sea and North Sea, were excluded (SNAP 8). The scenario run was subtracted from the base run to obtain the
impact of the international maritime sector in the Baltic Sea and the North Sea.
**3.2 Model performance of concentrations**

Particulate matter is difficult to measure and various measuring instruments register different types of particles, which result in some uncertainties to input data. Also, some semi-volatile compounds exist in both gaseous and particle form and the definition of the different particle groups vary in different countries. Moreover, there are still components of the coarser particles, such as aerosol and biogenic organic farming dust that are not included in EMEP model. Another uncertainty of the input data is that not all nations included in the EMEP area report yearly emissions (Gauss et al., 2015). We validated the EMEP modelled data for 2013 using measured concentrations of nitrogen dioxide, sulphur dioxide and particulate matter from stations Vavihill and Utö.

Measured data were compared to daily averages of modelled data from the 50 km $\times$ 50 km grid box where the measurement sites were localized. If measured data were specified in hourly values, calculations of daily averages were made. When measured data were missing for one day, the validation for this day was excluded in the evaluation. The evaluation included calculations of daily average, bias, correlation, root mean square error (RMSE) and also the P-test and scatter plots of model results versus measured data of the daily average concentrations of sulphur dioxide, nitrogen dioxide and particulate matter.

**3.3 Future Ship Emissions**

Five future scenarios differing with respect to the sulphur content of the fuel and scrubber usage of the shipping fleet were developed (Table 2). Scenario no. 1 corresponds to the fuel content regulation January 2010 to December 2014 (1% sulphur in the fuel), and scenario no. 3 (0.1% sulphur in the fuel) corresponds to the regulations from January 2015. Scenario no. 2 has been included since it in Sweden was suggested as an alternative, low-cost reduction in sulphur content (0.5% sulphur in the fuel). In scenario 4 and 5, use of open-loop wet scrubber technique for removing sulphur from the exhaust is assumed for 50 and 100% of the fleet respectively. The use of scrubber is assumed to increase linearly from no scrubber installations at all. The increase rate of the proportion of ships using scrubbers are the same for scenario 4 and 5 but ends at 50% or 100%, respectively. Hence, these are similar until 2020. It is assumed that the fuel used in the ships with scrubbers will have an average low-cost sulphur content of 2.7%, corresponding to the current average outside SECA (ENTEC, 2005). Further it is assumed that the ships in the basins north from Baltic proper, Archipelago, Åland and Bothnian Seas and Bothnian Bay, cannot use the scrubber technique to a large extent. This is due to the ice properties in the winter and the low alkalinity. For scenarios 4 and 5 the emissions to the atmosphere are estimated to correspond to 0.1% sulphur in the fuel (following the regulations). To achieve atmospheric emissions corresponding to 0.1% sulphur in the fuel it is assumed that 96% of the sulphur is taken up in the scrubber, the scrubber water is discharged untreated and the sulphur oxides are directly transformed into strong sulphuric acid. Regulations of nitrogen oxides emissions are in an early stage. Therefore, these emissions are assumed to increase at the same rate as the shipping traffic. We here follow the TREMOVE European transport model (De Ceuster et al., 2006), which gives an increase of 2.5% and 3.9% per year for cargo and passenger traffic, respectively. In addition, the effect of a NECA was studied, following assumptions made in J15. There is no seasonal variation in ship emissions in the ENTEC/IIASA inventory (2011) and the monthly variation through the years 2006–2009 presented by Jalkanen et al. (2014) is rather low ($\pm$10%). Therefore, no seasonal variation in the future emissions is assumed.

**Table 2.** Future scenarios that differ with respect to the sulphur content of the fuel and scrubber usage

| Scenario no. | Shipping not using wet scrubbers | | Shipping using wet scrubbers | |
| --- | --- | --- | --- | --- |
| | *% of total* | *% sulphur in fuel* | *% of total* | *% sulphur in fuel* |
| 1 | 100 | 1.0 | 0 | |
| 2 | 100 | 0.5 | 0 | |
| 3 | 100 | 0.1 | 0 | |
| 4 | 50 by 2020 | 0.1 | 50 by 2020 | 2.7 |
| 5 | 0 by 2025 | 0.1 | 100 by 2025 | 2.7 |

### 3.4 Deposition scenarios of ship emissions

In Omstedt et al. (2015) a database for ship emissions and the corresponding depositions was constructed for the 1900–2011 period using a combination of emission databases (ENTEC/IIASA, EDGAR 4.2 and EDGAR-HYDE 1.3) and deposition from the EMEP model. For the years 2006–2009, the emission distribution was scaled to correspond to that presented by Jalkanen et al. (2014); for 2010, linear interpolation was used. Ship traffic was also assumed to follow the regulated changes in fuel sulphur content in the SECA area. We assume 2.7% sulphur in the fuel until May 2006, 1.5% until the end of 2009, and 1% thereafter. This information was also used to correct the EDGAR 4.2 emissions fields. Further back in time, the emission fields from 1900 to 1970 from EDGAR-HYDE 1.3 were used.

We here extend the database into the future using the alternative scenarios described in Section 3.3. We also use a similar methodology as in Omstedt et al. (2015), with the reference year 2011. The spatial distribution of atmospheric deposition of sulphur oxides and nitrogen oxides from ship traffic was estimated by the EMEP model. The model was first run for the meteorological years 2009 to 2011 with emissions from 2011. The variation of deposition between the three years indicated that inter-annual effect of meteorology was low for annual deposition. Initial analysis, to help find the better method revealed that dry deposition is more focused along ship routes than wet deposition; the dry part of the deposition was thus assumed to be scaled by the local emissions. The wet deposition was more spread. Local ship emissions accounted for approximately 25% of deposition in the central Baltic Sea and approximately 45% of wet deposition in the Kattegat. In the Kattegat, almost half of the wet deposition originated from North Sea ship traffic, whereas a very small proportion of the wet deposition in the Baltic Sea east of Bornholm originated from the North Sea. Therefore, the wet deposition trends of sulphur oxides and nitrogen oxides, in each basin, was set equal to the local emission, except for Kattegat, Belt Seas and Øresund, where 50%, as a first-order approximation, was assumed to depend on emission trends in the North Sea. This approach resulted in a reference year (2011) of deposition-to-emission ratios with a monthly resolution. The relative seasonal variability was kept throughout the period.

Non-ship emission trends follow the RCP 4.5 scenarios from 2010 (Lamarque et al., 2010) and deposition simulations (Engardt and Langner, 2013) using the MATCH model (Robertson et al., 1999). Ship emisson from RCP 4.5 (Eyring et al., 2010), including the traffic distribution from ICOADS (Wang et al., 2008), was replaced by our scenarios described in Table 2. Total emissions were calculated by correspondingly correcting the MATCH output. Last, the spatial fields were averaged into the Baltic Sea basins defined in Fig. 2.

**4 Results**
**4.1 Ship deposition scenarios**
2011 is the base year for the deposition scenarios. That means that the spatial relative distribution of ship effects
remains the same during the scenarios. Fig. 3 shows the ship emissions and spatial distribution of dry, wet and
total deposition in 2011. Even if the ship emissions remain constant throughout the year, the weather causes the
deposition pattern to change. In Fig. 4 the seasonal variations (mean of the meteorological years 2009–2011 and
emissions from 2011) of the deposition of OXS are shown in six Baltic Sea basins (Kattegat, Arkona, Bornholm,
Eastern Gotland, G. of Finland and Bothnian Bay basins). In most basins there is a minimum during the summer
and a maximum in the autumn.
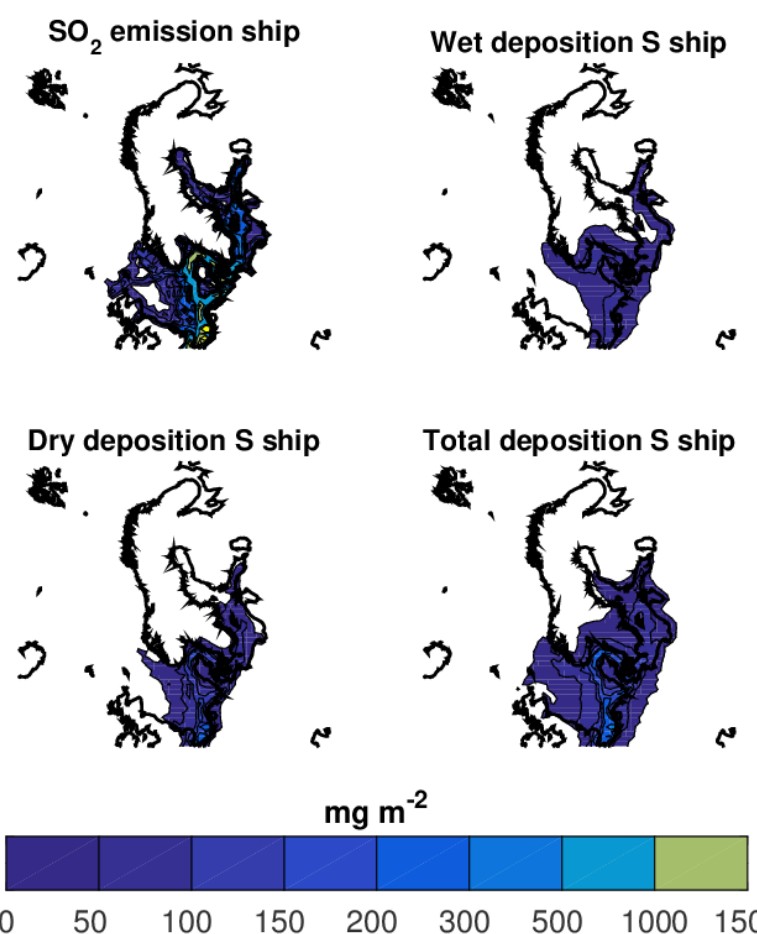
**Figure 3** Total emissions of SO$_x$ and deposition of OXS from international shipping in the Baltic Sea and North
Sea in 2011.

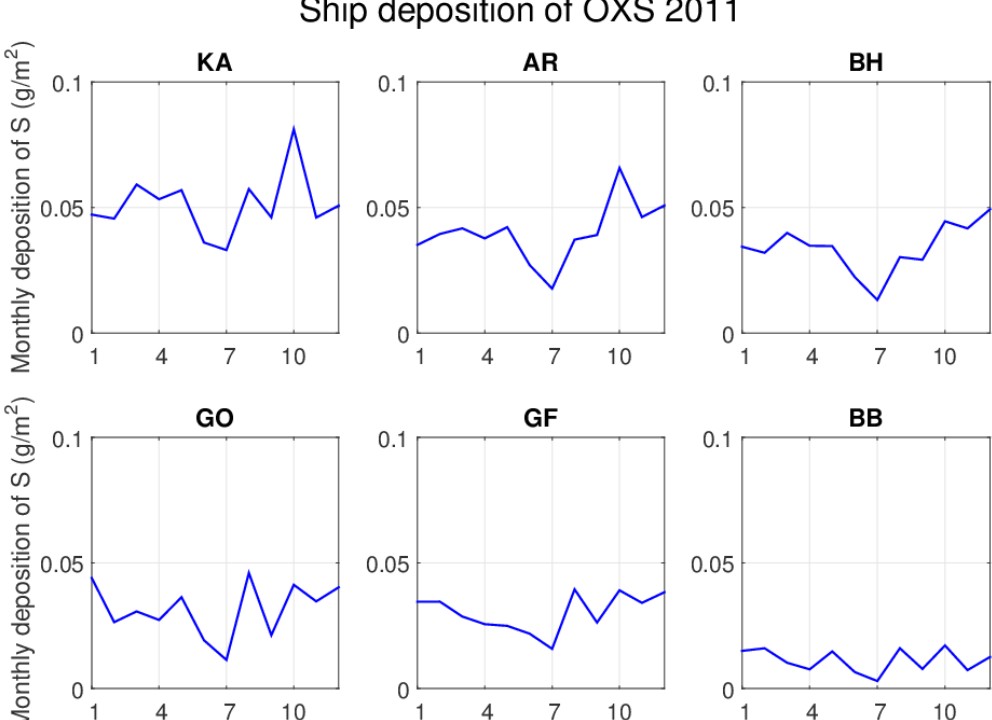

**Figure 4.** Monthly deposition of oxidized sulphur (OXS) in six basins of Baltic Sea (defined in Fig. 2).

The ship emissions scenarios in Baltic Sea are shown in Fig. 5. Emissions into the atmosphere (Fig 5a) all show positive trends after 2015, due to the increasing traffic. With the traffic scenarios emissions have more than doubled in 2050 compared to 2015. The reduction of sulphur content in fuel will, as expected, result in a reduction in sulphur emissions into the atmosphere in the Baltic Sea area. With scrubber, the OXS goes directly into the water along the shipping lines. Note that it is assumed that ships are not using scrubbers north of Baltic proper (as discussed in section 3.3). Averaged over the whole complete Baltic Sea (Fig 5b) it is seen that, if using 2.7% sulphur fuel, the input of OXS into the sea is trespassing the deposition from the 1% scenario already by 2020. This is regardless of which scenario (4 or 5) is used. If all ships in the region south of Åland are using scrubbers, and fuel with a sulphur content of 2.7%, the emission of sulphur oxides into the Baltic Sea is expected to be almost three times the size in 2050, compared to if no scrubbers were used and fuel with sulphur content of 1.0% (Fig. 5b).

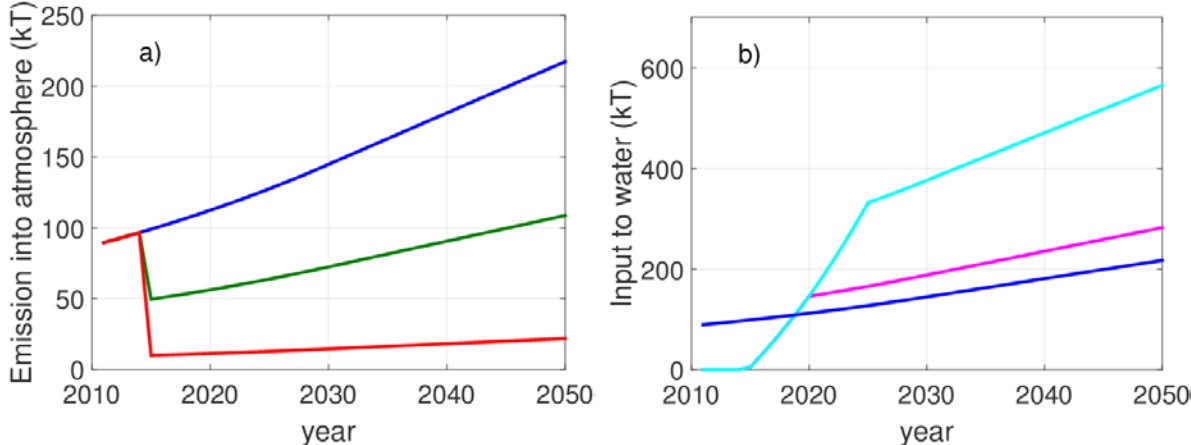

**Figure 5.** (a) Emissions of oxidized sulphur into the atmosphere in the Baltic Sea area (kT) in 2010 to 2050 for
Scenario 1 to 3. The blue line corresponds to Scenario 1, the green line to Scenario 2 and the red line to Scenario
3, (b) Emission directly into the water of the Baltic Sea (kT) for scenario 4 to 5 and 1 in 2010 to 2050. The magenta
line corresponds to Scenario 4, the cyan line to Scenario 5. For comparison the blue line shows the atmospheric
deposition from scenario 1.
The deposition scenarios of sulphur from shipping, together with the historical data from Omstedt et al. (2015) are
shown in Fig 6. The deposition of sulphur from ship emissions in the Baltic Sea increased rapidly until the 1970s
and then more slowly until 2005 (e.g. Claremar et al., 2013). Applying scenario 2 or 3 the deposition becomes
significantly lower. The total deposition of sulphur in the Baltic Sea, from all emission sources, reached its
maximum in the second part of the 1900s. It has decreased steadily since then and the deposition of sulphur is
expected to continue to be low for the examined time period from present to 2050 (Fig. 7). The contribution of
deposition of oxidized sulphur from shipping is expected to increase somewhat from 2010 to 2050 in all basins of
the Baltic, but the levels will stay at low levels. The deposition of sulphur from all emission sources is predicted
to be rather invariable from 2010 to 2050, as given from RCP4.5.

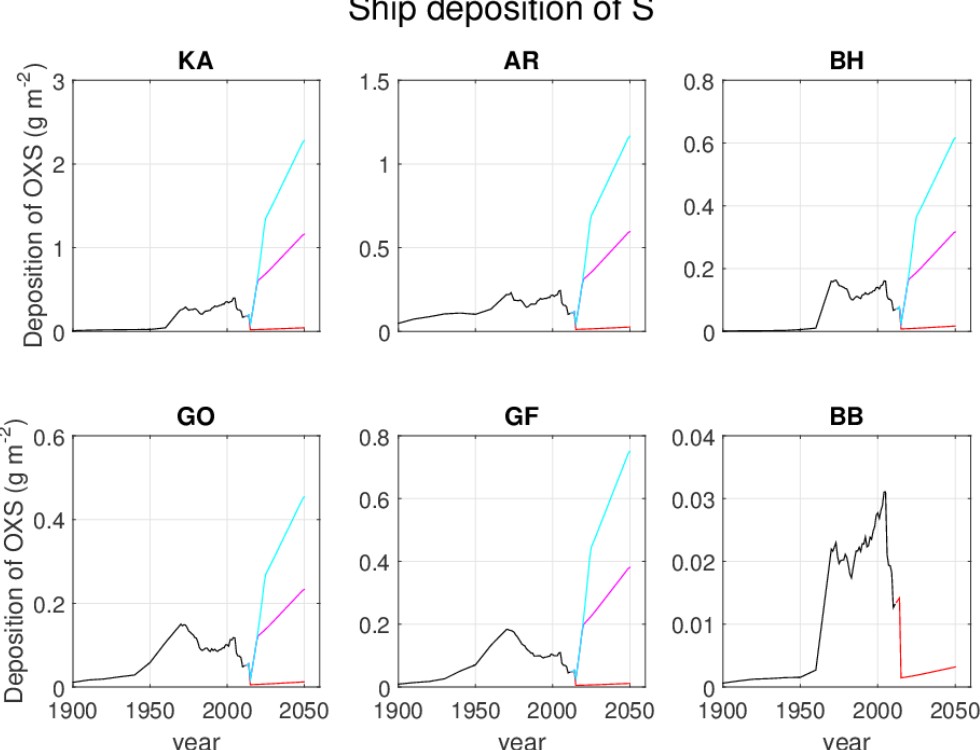

**Figure 6.** Annual ship deposition of sulphur (mgm$^{-2}$) in six basins of Baltic Sea (defined in Fig. 2), year 1900 to 2050. The red line corresponds to Shipping scenario 3, the magenta and cyan line to Shipping scenario 4 and 5, respectively (scrubber + atmospheric deposition). Black line to historical shipping (derived in Omstedt et al., 2015). It is here assumed that scrubbers are not used in Bothnian Bay.

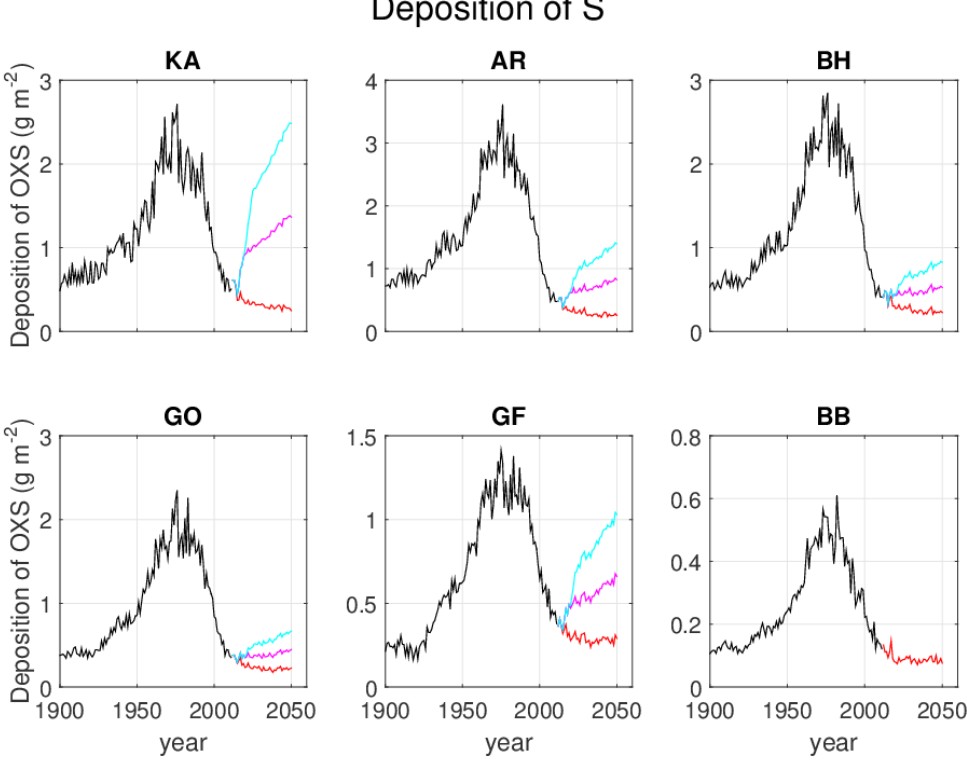

**Figure 7**. Annual deposition of sulphur from all emission sources (gm$^{-2}$) in six basins of Baltic Sea (defined in
Fig. 2), year 1900 to 2050. The red line corresponds to Shipping scenario 3, the magenta and cyan line to Shipping
scenario 4 and 5, respectively (scrubber + atmospheric deposition). The black line shows historical shipping
(derived in Omstedt et al., 2015).
The deposition in the whole Baltic Sea is presented in Table 3. The ship contribution to the total atmospheric OXS
deposition decreases, from 18% in 2011 to 5% in 2030 and 7% in 2050 (scen. 4). With scrubber, the contribution
adds up to more than 50%. In 2050 the atmospheric deposition has been reduced from 160 to 95 Mg yr$^{-1}$ but with
scrubber the input into the Baltic Sea is increased to 227 Mg yr$^{-1}$, an increase by 42% (scen. 4).
**Table 3.** Total deposition of OXS and OXN in Baltic Sea Mg yr$^{-1}$.

|  | OXS | | OXN | |
|---|---|---|---|---|
|  | w/o scrubber (scen. 3) | w/ scrubber (scen. 4) | w/o NECA | w/ NECA |
| Tot dep 2011 | 160 | 160 | 77 | 77 |
| Tot dep 2030 | 94 | 183 | 71 | 52 |
| Tot dep 2050 | 95 | 227 | 88 | 37 |
| From ships 2011 (%) | 18 | 18 | 35 | 35 |

| | | | | |
|---|---|---|---|---|
| From ships 2030 (%) | 5 | 51 | 60 | 45 |
| From ships 2050 (%) | 7 | 61 | 72 | 34 |

The deposition of nitrogen from ship emissions is expected to increase to all the basins in the Baltic Sea from present to 2050 as we do not include any coming regulations on nitrogen (Fig. 8). The increase is due to increase in traffic scenario. The total deposition of nitrogen in the Baltic Sea, from all emission sources, is expected to increase in the Baltic Sea compared to current deposition level (Fig. 9). The increase of nitrogen deposition varies significantly for the different basins and for the Kattegat basin the highest values of nitrogen deposition in the 1970s will be exceeded before year 2050. The contribution of deposition of oxidized nitrogen from shipping is expected to become a more significant contributor to total deposition of oxidized nitrogen from 2010 to 2050 in all basins of the Baltic Sea (Table 3). The OXN deposition is significantly lower than in J15. They used the EC4MACS Interim Assessment (Amann et al. 2011) which indicates that the RCP4.5 has lower scenario on nitrogen. That means that if using EC4MACS data, ship part of OXN deposition would be smaller, but total effect be larger.

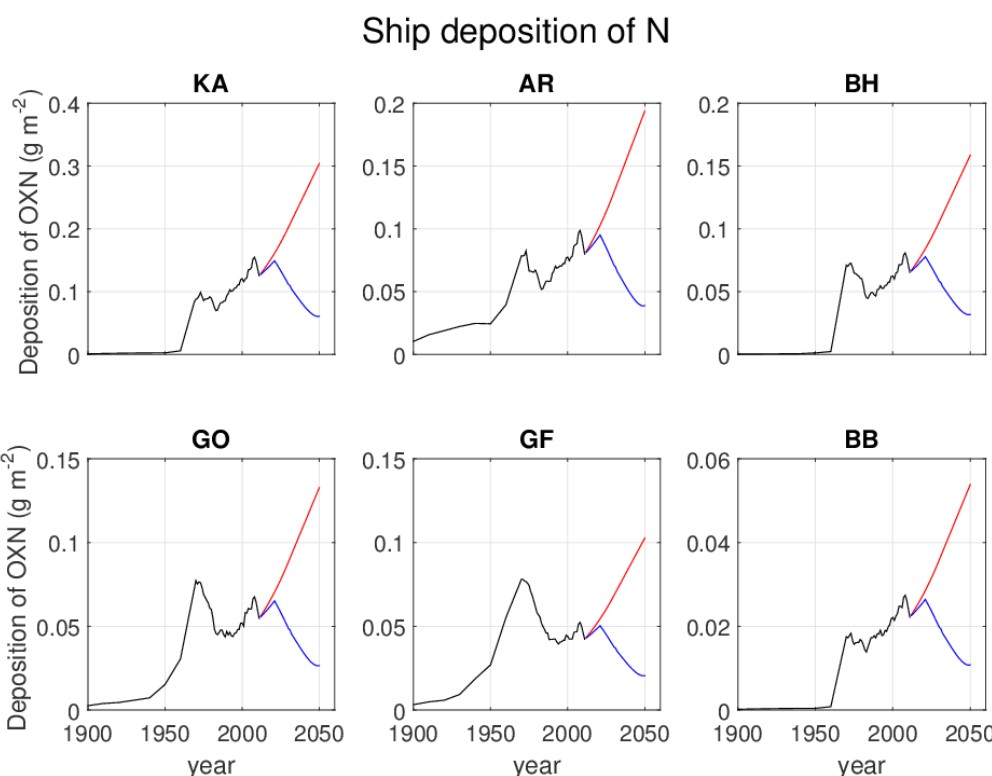

**Figure 8**. Annual ship deposition of nitrogen (gm$^{-2}$) in six basins of Baltic Sea (defined in Fig. 2), year 1900 to 2050. The red line corresponds to Shipping scenario 1 to 3 and the black line to historical shipping (derived in Omstedt et al., 2015). Blue line is with implementation of TIERII and NECA.

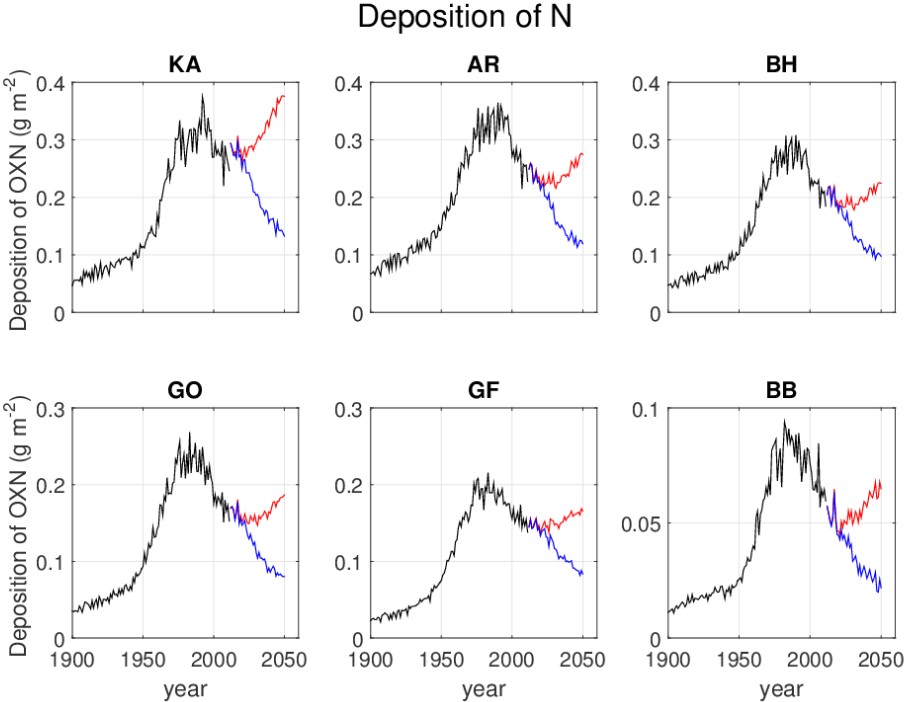

**Figure 9.** Annual deposition of nitrogen from all emission sources (gm$^{-2}$) in six basins of Baltic Sea (defined in Fig. 2), year 1900 to 2050. The red line corresponds to Shipping scenarios 1 to 3 and the black line shows historical shipping (derived in Omstedt et al., 2015).

The deposition for OXN in the whole Baltic Sea is presented in Table 3. The ship contribution to total increases for the atmospheric deposition, from 35% in 2011 to 60% in 2030 and 72% in 2050. In 2050 the atmospheric deposition has been reduced from 77 to 88 Mg yr$^{-1}$, not counting for NECA.

We have, so far, not accounted for the long-term shift to TIER II and TIER III in NECA. A decision of a NECA in Baltic Sea and North Sea was taken while preparing this paper. TIER II was introduced in 2011 and TIER III will be introduced in 2021. In Table 4 estimates from J15 are shown and a reduction of emissions of 26% in Baltic Sea and 29% in North Sea can be concluded in 2030, relative to without a NECA, i.e. to ships, mainly following TIER II. We apply on our deposition a reducing factor due to both TIER II implementation and the NECA in 2021. We assume that half of the fleet has implemented each regulation after 10 years, and completely after 30 years. This yields power curves with exponent 1.7 as long as there are not more than three concurrent fleets. Further we assume that TIER I was completely implemented in 2012 and that the implementation for TIER I is replaced by TIER II until 2021 when TIER III takes over. Hence, in the period 2021-2031 when all TIER fleets are present, the TIER II fleet remains constant at 50% until 2031 when TIER III reaches 50%. After that TIER II is the only remainder to the TIER II fleet. The estimated fleet parts are presented in Fig. 10a. The ship emissions of NOx in the Baltic Sea is then scaled, using fleet weighted factors, with 85% for TIER II and 20% for TIER III, relative to TIER I (J15). The resulting factors are shown in Fig. 10b. The deposition from shipping is then assumed to follow the trends of emissions in the NECA in the basins, shown in Figs. 8, 9 (blue line). The deposition for OXN in the whole Baltic Sea with the effect from NECA and TIER II is presented in Table 3. The ship contribution to total

first increases for the atmospheric deposition, from 35% in 2011 to 45% in 2030 and then decreases to 34% in
2050. In 2050 the atmospheric deposition has been reduced from 77 to 37 Mg yr$^{-1}$.
**Table 4.** Emissions from shipping in the Baltic Sea in Mg yr$^{-1}$.

|  | SO2 | | | | NO2 | | | |
|---|---|---|---|---|---|---|---|---|
|  | BS | | NS | | BS | | NS | |
|  | This study | J15 | This study | J15 | This study | J15 | This study | J15 |
| 2011 rv.4.4 and rv.4.8 | 89 | 80 | 201 | 155 | 304 | 337 | 678 | 662 |
| 2012 | 73 | - | 165 | - | 313 | - | 700 | - |
| 2013 | 62 | - | 140 | - | 247 | - | 555 | - |
| 2030 scenario 3, w/o TIER II-III | c. 15 | 8 | 34 | 21 | 492 | 293 | 1097 | 642 |
| 2030 scenario 3, w/ TIER II-III | c. 15 | 8 | 34 | 21 | 264 | 217 | 589 | 457 |

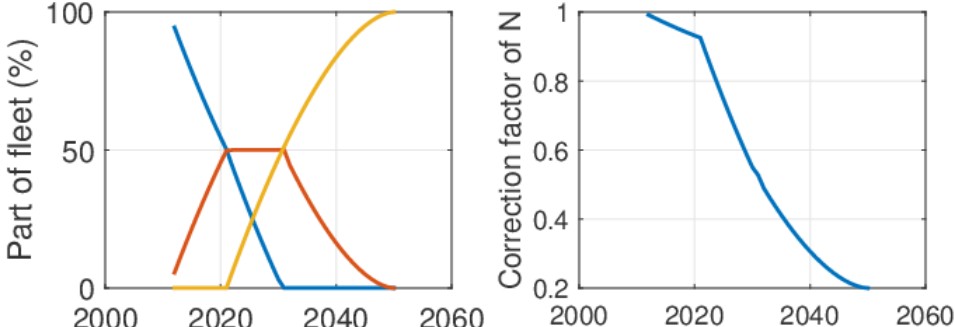

**Figure 10.** In a) estimated part of fleet applying to TIER I (blue), TIER II (red) and TIER III/NECA (orange), in
b) correction factor for OXN deposition from shipping, using the implementation of TIER II and the TIER III in
NECA from 2021.
The deposition of OXS and OXN together acts as strong acids in the water. The result is an acidifying effect and
a pH decrease (Fig. 11). At pH 8, a proton input of 1 nmol m$^{-2}$ corresponds to a decrease in pH of $4 \cdot 10^{-6}$ for a
mixed ocean surface layer of 10 m. For pH 8.2, this number is $7 \cdot 10^{-6}$. The largest effect is seen in Kattegat where
ship traffic is high. In the worst case scenario, even with NECA, proton input is as high in 2050 as in 1970 to 1990.
In the larger basins in Baltic Proper, e.g. Eastern Gotland basin and Bornholm basin, the effect is smaller, whereas
in Gulf of Finland the proton input is almost as high as in 1970 to 1990. It is concluded that the introduction of
NECA really lowers the nitrogen input into Baltic Sea and the acidification is limited. However, using scrubbers,
the effect is limited to a few percent in proton input.

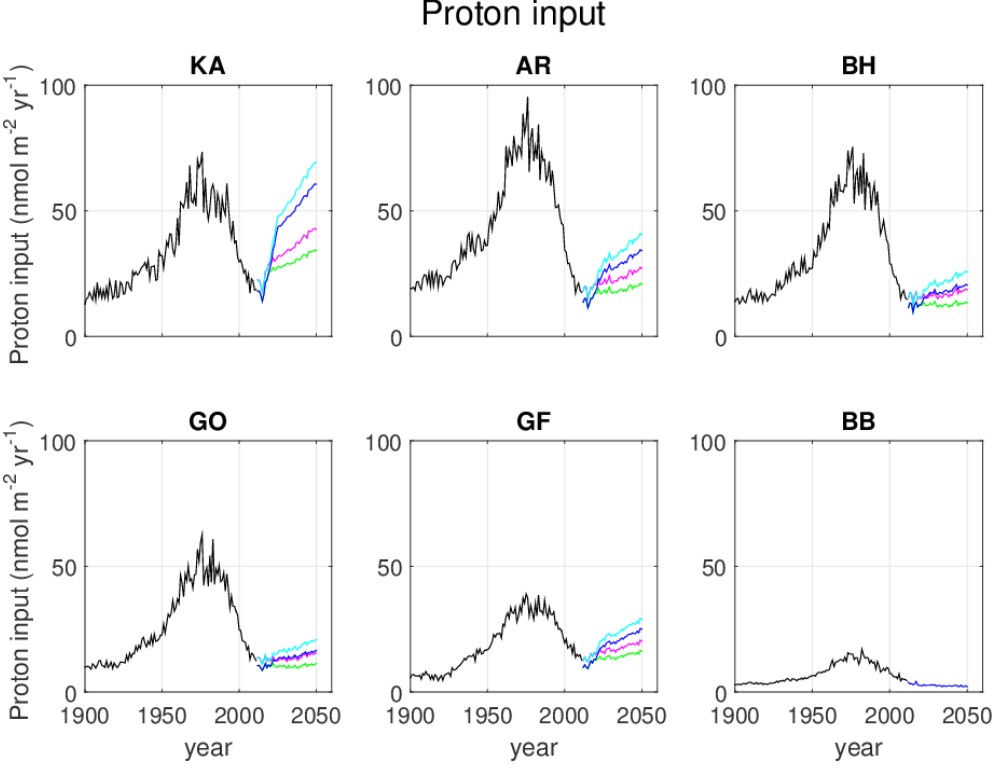

Figure 11. Annual proton input from OXS and OXN in six basins of Baltic Sea (defined in Fig. 2) for year 2010 to 2050. The red line corresponds to Shipping scenario 3, the magenta and cyan line to Shipping scenario 4 and 5, respectively (scrubber + atmospheric deposition). The black line shows historical shipping (derived in Omstedt et al., 2015). Green and blue lines is with implantation of TIER II and NECA and scrubber scenario 4 and 5, respectively.

The difference in physics in EMEP model rv.4.4 and 4.8 causes different deposition. With the assumption that rv.4.8 better represents the reality a correction was calculated for the Baltic Sea basins. The result is shown in Fig. 12 for OXS and OXN. It is seen that for OXS the difference is below 5%, and with a decreasing trend counting from Kattegat to Bothnian Bay. The pattern for OXN is almost the opposite with the largest correction in the northern parts. Also, the correction is higher, 10 to 30%. The impact on the future scenarios is discussed in next section. The implication of this is that this has very small effect on sulphur, because the atmospheric deposition is already low, especially compared to the exhaust from the scrubbers. The nitrogen is more important, although the largest relative effects of the correction is in the Bothnian Bay where the deposition is smallest, just 50 mg $m^2$ in 2050. Therefore, and since the scenarios are connected to other uncertainties, a correction was not made.

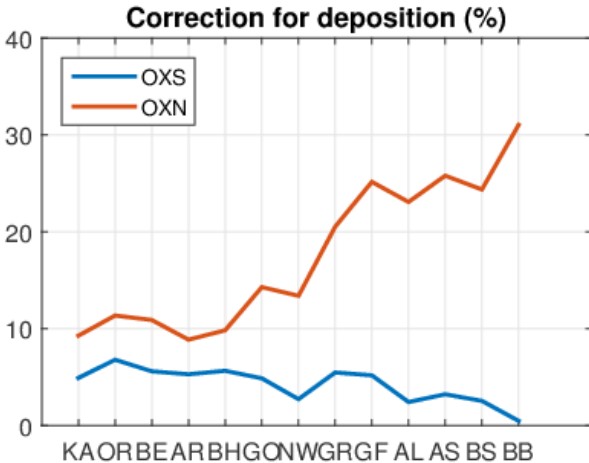

**Figure 12**. Correction of deposition caused by ship traffic, in the different Baltic Sea basins.
**4.2 Present emissions and surface concentrations**
The spatial distribution of the emission from international shipping in the Baltic and North Sea in 2011 (from
EMEP model) is demonstrated in Fig 3a. The emission distribution (relative) is almost the same in 2013. The
highest emission levels of the pollutant are found near big ports and shipping lanes, especially in the area around
the English Channel and Denmark. Compared to ship emission 2011 there is a decrease in the inventory used for
the EMEP model (Table 4). There has been a decrease although not stepwise as if all ships were using 1% sulphur
in fuel, directly. This may be an effect of the interpolation as mentioned in section 2.3 (Fagerli et al., 2015), i.e. an
underestimation of the real ship emissions in 2012 and 2013. In Table 4 also the emissions used in J15 are shown.
There are sometimes large differences, possibly an effect of the resolution of data and Baltic Sea basin areas. The
largest deviation is for North Sea, but this area is not directly analysed in the present investigation.
The evaluation of the EMEP model concentrations (with rv.4.8) in 2013 is summarized in Figs. A1 and A2 and
Table A1 in the Appendix. The yearly averages of the measured and modelled concentrations of the pollutants
were rather consistent (Table A1). The EMEP model underestimates the concentrations of $NO_x$, $SO_2$ and PM at
both Utö and Vavihill, except for $NO_x$ at Vavihill. The underestimations may be an effect of the underestimated
emissions mentioned above (Fagerli et al., 2015). The model has some difficulty to model the maximum values of
the observed data (Fig. A1), possibly an effect of the resolution.
The seasonal variability of modelled and measured concentrations at Vavihill and Utö in 2013 is shown as monthly
averages in Fig A2. There is an overall good agreement for most of the pollutants. However, $NO_2$ and $SO_2$ at Utö
deviated significantly for some time periods and $PM_{10}$ at Vavihill (Fig. A2, g). The seasonal changes are well
captured, but the variability is rather underestimated by the model. Bias, mentioned above, is also evident here.
An aspect of the evaluation is that observed data from point measurements were compared modelled data from
gridded boxes with the size of 50 km × 50 km. The regional resolution of the model, results in loss of variations
in the grid box and an average for the entire grid box is calculated, which in this study may have resulted in an
underestimation of the maximum values in shipping lanes and ports. This may also be a reason to why the model,
in general, had some difficulty to model the maximum values.
WHO guidelines for the annual average exposure for $PM_{2.5}$ = 10 $\mu gm^{-3}$; $PM_{10}$ = 20 $\mu gm^{-3}$; $NO_2$ = 40 $\mu gm^{-3}$; $SO_2$
= 20 $\mu gm^{-3}$ (World Health Organization, 2006). EMEP model calculations by J15 have shown that in 2010 the
WHO guidelines of the annual averages for $PM_{10}$ and $PM_{2.5}$ are exceeded in parts of the EMEP area. In the present
calculations for the year 2013 the concentrations of particulate matter still exceed the WHO guidelines in some
restricted parts of the Baltic Sea area (Fig. 13). Surface concentrations (near surface concentrations at 3 m) of $SO_2$
and $NO_2$ do not exceed WHO guidelines in 2011 to 2013 according to our EMEP calculations. It is also seen that
concentrations decreased between 2011 and 2013, as an effect of less ship emissions in the EMEP model input.
The $PM_{2.5}$ concentration is in-line with J15 but $PM_{10}$ is much higher in the present study over the North Sea,
probably because of different definition of the content. The explanation cannot be referred to sea salt, which is on
one order of magnitude smaller.

**PM2.5 rv4.4**          **PM10 rv4.4**

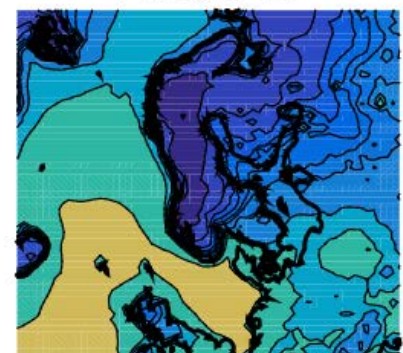

**PM2.5 rv4.8**          **PM10 rv4.8**

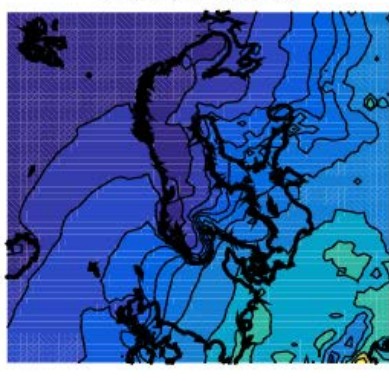          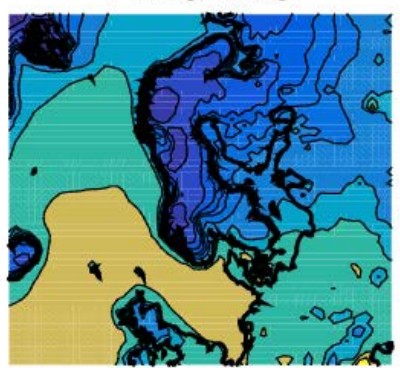

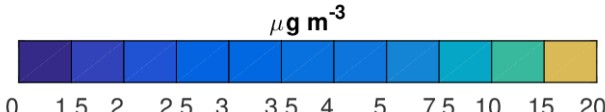

**Figure 13.** Annual mean concentration in 2011 of near-surface concentration (at 3 m level) of particulate matter
from all emission sources in the EMEP area.
International shipping in the Baltic Sea and the North Sea contributes significantly to total surface concentration
of nitrogen oxides, sulphur dioxide and particular matter in 2009 to 2013. In some areas in the Baltic Sea region,
the contribution of nitric oxide, nitrogen dioxide and sulphur dioxide from international shipping represent up to
80% of total concentration of the pollutants from all emissions sources in 2013. For $PM_{2.5}$, the contribution from
shipping to total concentration was a maximum of around 20% and, for $PM_{10}$, 13%. The highest concentrations of
the pollutants are found near big ports and shipping lanes, where the shipping activities were most intense (Fig.
14). The highest concentrations of nitric oxide, nitrogen dioxide and sulphur dioxide are more clearly along the
shipping lanes, compared to $PM_{2.5}$ and $PM_{10}$, in agreement with Aardenne et al. (2013). Variations in the results
between 2011 and 2013 are due to weather pattern and the decrease in ship emissions. Hence, locally the difference
between 2013 and 2011 can be up to 10 percentage units, both up and down, but less for PM, on the order of 2
percentage units. The actual annual mean concentration of PM2.5 from shipping is shown in Fig. 15. Comparing
to J15, the values and spatial distribution is overall good. However, there is a tendency of smearing out, expected
from the lower resolution in our study.

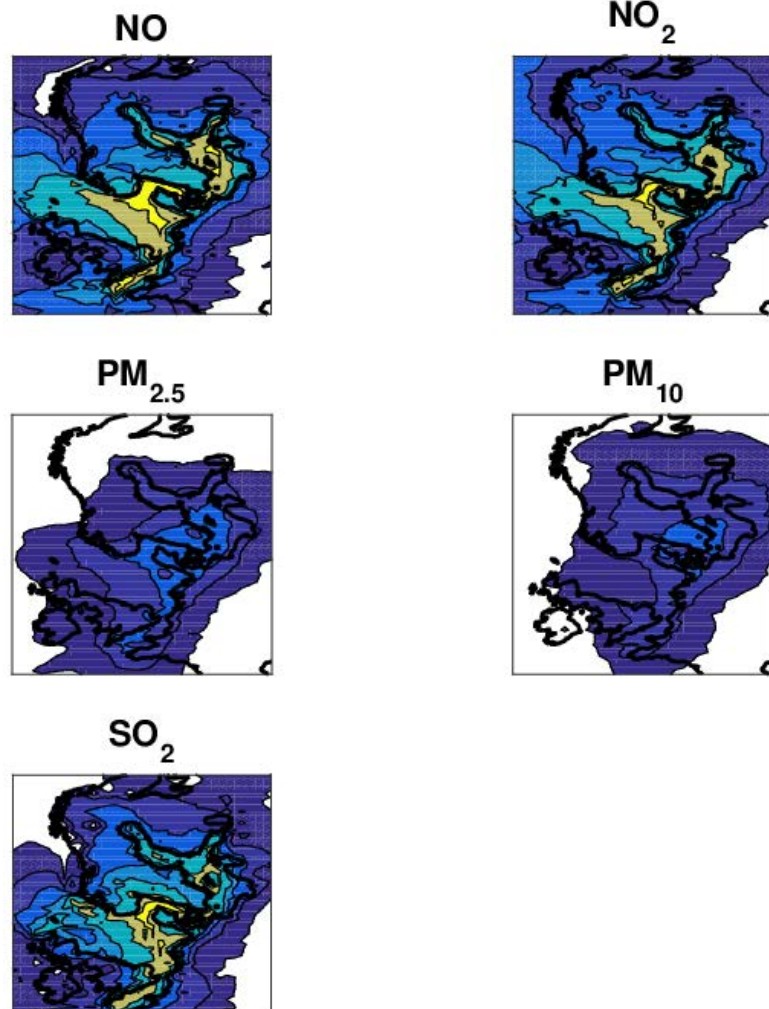

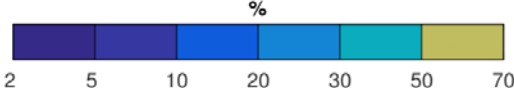

**Figure 14.** Percentage (%) of the total surface concentration, caused by international shipping in the Baltic Sea
and the North Sea in 2011.

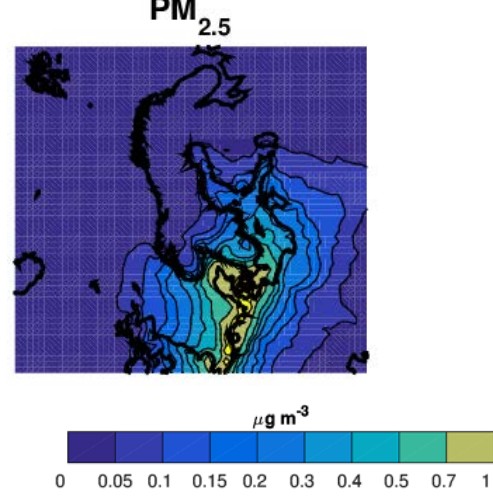

**Figure 15.** Concentration of $PM_{2.5}$ caused by shipping.
**4.3 Present Deposition**
The simulated (by rv.4.8) wet deposition in 2013 from shipping in the Baltic Sea and North Sea reach over 60
$mg(S)m^{-2}$ and 80 $mg(N)m^{-2}$ in some of the areas in Europe (Fig. 3d and 17). The amount of wet deposition of the
pollutants is high in coastal areas, which may be due to enhanced precipitation by coastal, orographic and frictional
effects on the meteorology.. This results in more deposited pollutants in countries with a long coastline. This is
consistent with the study of J15 where it was found that the deposition of nitrogen from shipping was high in the
seas and at coastlines. The large areas of OXN deposition over southern Norway, west coast of Sweden, and west
of Norway are both seen here and in J15. The dry deposition for the same year reach as maximum over 200
$mg(S)m^{-2}$ and 65 $mg(N)m^{-2}$. The highest cumulative wet and dry depositions are found in areas close to some of
the shorelines in Europe and near big ports and shipping lanes (cf. Fig. 3). The total (wet and dry) cumulative
deposition of oxidized sulphur reached high values along the shipping lanes and its maximum values are found in
areas around the inlet to the English Channel. The maximum values of the total (wet and dry) cumulative deposition
of oxidized nitrogen are found at the Swedish west coast. Numbers and patterns of the N deposition are in-line
with J15. This indicates that the resolution higher than 50 x 50 $km^2$ is not of major importance for deposition of
the basin scale. Variations in the results between 2011 and 2013 are due to weather pattern and the decrease in
ship emissions. Hence, locally the difference between 2013 and 2011 could be up to 10 percentage units, both up
and down.

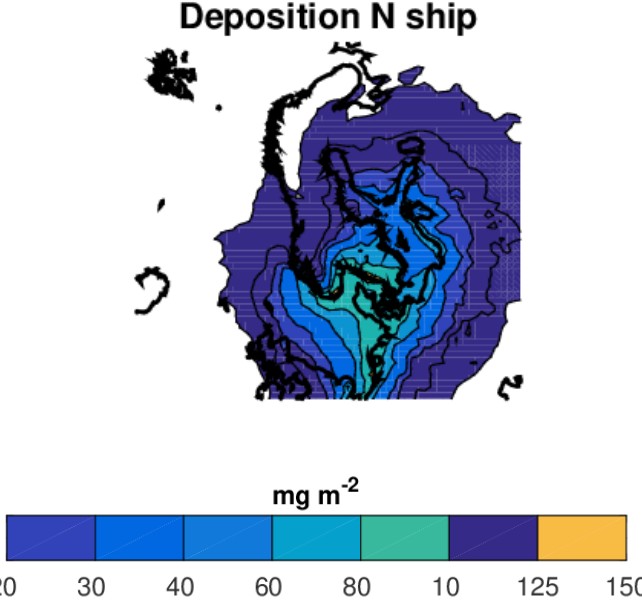

**Deposition N ship**

mg m$^{-2}$

| 20 | 30 | 40 | 60 | 80 | 10 | 125 | 150 |

**Figure 16.** Deposition of OXN caused by shipping.

International shipping in the Baltic Sea and North Sea contributes significantly to the deposition of oxidized sulphur and nitrogen, in 2009 to 2013. In 2013 the percentage contribution from the shipping to the total cumulative wet deposition of sulphur from all emissions sources reaches 29% in some areas of the Baltic Sea region and the contribution of dry deposition of sulphur is calculated to a maximum of 84% of total dry deposition of sulphur in (Fig. 17). The percentage contribution of wet deposition of nitrogen reaches a maximum of 28% and the contribution of dry deposition of nitrogen reached a maximum of 47%. Contribution of ship emissions to the total (wet and dry) annual deposition of sulphur is as much as 56% in some areas and for nitrogen 29%. Deposition pattern for the dry and wet deposition of oxidized sulphur and nitrogen differs slightly when wet deposition is spread over a larger area than dry deposition. Dry deposition is more focused along ship routes. Dry deposition of the pollutants caused by shipping represents, on the other hand, a higher percentage of total amounts of the deposition than the wet deposition from shipping. The percentage contribution of dry deposition from shipping is higher for oxidized sulphur than oxidized nitrogen.

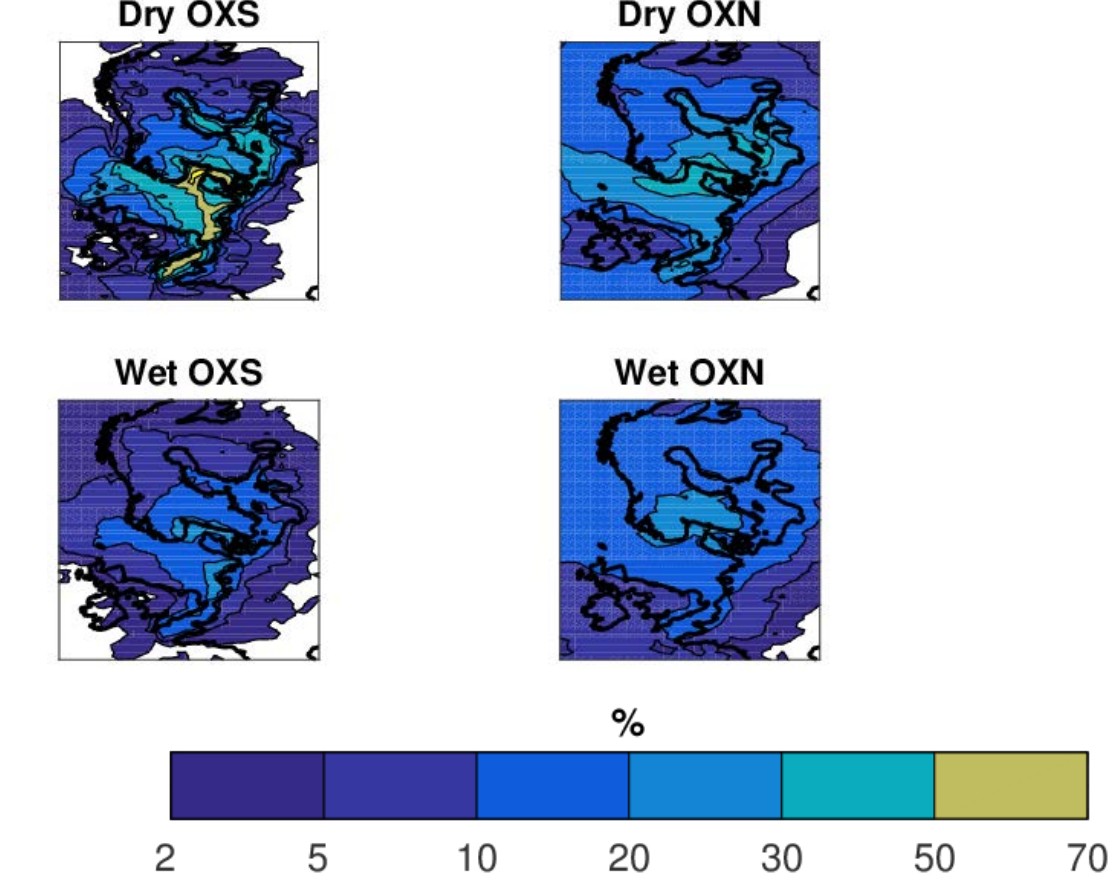

**Figure 17.** Percentage (%) of the deposition, caused by international shipping in the Baltic Sea and the North Sea in 2013 of (a) Dry OXN, (b) Dry OXS, (c) Wet OXN, (d) wet OXS.

## 5 Discussion

We have in this investigation focused on the impact from scrubbers in the future on sulphur deposition, its potential acidification of the Baltic Sea, and in addition also included oxidized nitrogen. We have not taken into account input from non-sea salt base cations, like calcium from cement industry, and ammonium from for instance agriculture. Trends in those may alter acidification. For instance, calcium emissions have decreased, at least in the 90s and 2000s (e.g. Claremar et al. 2013). Similar deposition scenarios wereused in Turner et al. (2017), but here reduced NOx emissions from the ship traffic is also evaluated. We here included estimations of the effect from TIER II and NECA from 2030. The conclusion that scrubbers increase the ocean acidification still holds, but it is decreased by less than 20 %, when including the effect from NECA. Without scrubber, the impact from NECA is very large on reduced acidification In other words, scrubbers offset the benefits of NECA. The introduction of 0.5% sulphur in fuel outside SECA is estimated to have a minor effect on the Baltic Sea deposition since the atmospheric deposition is as low already now. The acid deposition from the scrubbers will also locally probably be, even if it is horizontally mixed, a magnitude, or more, larger than the basin averages. In the worst case scenario, at the basin scale, and assuming a mean mixing depth of 10 m of the surface water, the pH decrease in Kattegat can be on the order $3 \cdot 10^{-4}$ per year (at pH 8.1). Locally, along shipping lanes, the pH decrease can be comparable to $CO_2$ uptake (Fig. 12a).

This modelling study was based on international shipping emissions, which means that the contribution of emissions from all shipping, including national, in the Baltic Sea and North Sea are somewhat higher than these results show. In further work it would be of interest to include national emissions. To obtain more robust results, national reported input data should be put under more control and a future study should as a suggestion also examine how much impact it has on the result that several countries do not give complete reports of their annual emissions, to reduce uncertainties in the model.

In this investigation we were using a mixture of modelling, and statistics, with all its uncertainties. We used constant meteorology in the future scenario, but limited the uncertainties by using average of three meteorological years (2009–2011). But in order to obtain more robust results a study over a longer period of time is required, in the best case meteorology for every year. The resolution of the model, 50 x 50 km$^2$, compared to in J15 14 x 14 km$^2$, was shown to have a minor effect on basin scale studies. However, to see local effects at the coast, finer resolution is needed. The validation to observations of coastal concentrations showed that the model performed well, given the rather low resolution.

The new regulation of permitted weight percentage of sulphur in marine fuel was introduced in January 2015, which makes it of interest to include 2015 and the following years in further studies, to analyse the outcome of the new regulation. The 2015 0.1% limit was implemented in the scenarios but it remains to validate the compliance and the traffic scenarios on the longer time scale. In further studies, it would also be of interest to include a validation study of the deposition of the pollutants. Scenarios are based on assumptions on shipping activities as well as fuel use and cleaning patterns. Alternative fuel or cleaning techniques might be developed giving alternative scenarios.

To identify the dispersion of the different components of particulate matter from shipping it would be of interest to model each component separately. Stricter regulations of sulphur content in maritime fuel and increased use of other fuels will result in a new mixture of particulate matter from shipping. This ongoing change of composition of the pollutants makes it of interest to understand the dispersion of each separate component. In further studies a better resolution of the model is recommended to be used to examine the impact on local level. No seasonal variations have been taken into account in this study. Results of the study of J15 demonstrate that emissions from the international shipping vary to some extent over the year. The seasonal cycle of acidifying deposition is of importance for the surface water, due to the biological cycle and stratification of the water, as seen in Omstedt et al. (2015).

**6 Summary and Conclusions**

Model calculations using the chemical transport model EMEP show that the shipping in the Baltic Sea and North Sea is an important source to high near-surface concentrations of nitric oxide, nitrogen dioxide, sulphur dioxide and particular matter, and deposition of oxidized nitrogen and sulphur in the Baltic Sea and North Sea area. The highest concentrations of the pollutants were found near big ports and along shipping lanes. There,the international shipping in the Baltic Sea and North Sea was responsible for up to 80% of near surface concentrations of nitric oxide, nitrogen dioxide and sulphur dioxide in 2013. For PM$_{2.5}$, the contribution from shipping to total concentration was up to 20% and, for PM$_{10}$, up to 13%. The. It can also be seen that the contribution from shipping is of importance also over larger areas at sea and over land where many people are exposed. The percentage contribution from international shipping to dry deposition of sulphur was calculated to a maximum of 84% and

contributions of dry deposition of nitrogen reached a maximum of 47% in 2013. Wet deposition from shipping was spread over a larger area than dry deposition. Dry deposition of the pollutants caused by shipping represented a higher percentage of total amounts of the deposition than the wet deposition.

Future scenarios of ship emissions and the use of open-loop scrubbers were combined with modelled deposition from ships and other sources forming scenarios of acid deposition in the Baltic Sea basins. The impact of the different scenarios differs slightly for the different basins in the Baltic Sea, with highest acidification in the southern basins. The scrubbers focus the sulphur along the shipping lanes. Ship part of acidifying ocean deposition increases for sulphur when including the scrubber water and for nitrogen oxides due to increasing ship traffic. Direct acidification of ocean deposition from shipping increases for sulphur when including open-loop scrubbers. The impact is even larger for the Baltic Sea as a whole, since almost all sulphur goes into the water and not to the surrounding land areas. The estimates of the reduction in oxidized nitrogen deposition from introducing NECA in 2021, showed that there may be a large reduction of acidification. But in relation to the worst case scenario with 100% scrubbers in 2050, this effect is minor.

Considering the negative effects of the studied air pollutants and as the pollutants are a contributing factor of several current challenges in the Baltic Sea and North Sea area, this study shows that continued analysis of the maritime sector is required, in order to achieve sustainable shipping in the Baltic Sea and North Sea. For the marine environment, a large-scale usage of open-loop scrubbers should be avoided, at least with the use of residual oil.

To conclude

- Open-loop scrubbers concentrate sulphur input along shipping lines, with enhanced potential for acidification, even if the atmospheric deposition is estimated to be low.
- Acidification from a fleet with 100% scrubbers, using high sulphur content fuel, may reach the total deposition levels along the shipping lanes from the 1970s to 1990s.
- Open-loop scrubbers will give a significant contribution to the marine environment, unless medium sulphur content distillates are used.

**Acknowledgements**

This work is part of the Swedish Research Council Formas programme "Commercial shipping as a source of acidification in the Baltic Sea (SHIpH)", contract no. 2012-2120. The SHIpH group is acknowledged for valuable discussion concerning the traffic and scrubber scenarios. It is also part of the "Baltic Earth – Earth System Sciences for the Baltic Sea Region" program. The SHIpH group is gratefully acknowledged for valuable discussions on the scenario definitions. Magnuz Engardt at SMHI is thanked for the MATCH model deposition output. EMEP/MSC-W is acknowledged for the EMEP model code and input data. The EMEP model computations were performed using resources provided by SNIC via the Uppsala Multidisciplinary Center for Advanced Computational Science (UPPMAX) under projects snic2014-1-75 and snic2015-6-139.

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

**Appendix: Concentration validation at Vavihill and Utö**
**Table A1**. Comparison of model daily concentration average results from the EMEP model and measured data for
2013 at Utö and Vavihill. Obs. = observed data, Mod. = modelled data, Corr. = correlation coefficient and RMSE
= root mean square error

| Station | Component | Obs. ($\mu gm^{-3}$) | Mod. ($\mu gm^{-3}$) | Bias (%) | Corr. (r) | RMSE |
|---|---|---|---|---|---|---|
| Vavihill | | | | | | |
| | $NO_2$ | 3.69 | 5.05 | 36.7 | 0.72 | 3.03 |
| | $SO_2$ | 0.42 | 0.38 | -8.2 | 0.70 | 0.35 |
| | $PM_{2.5}$ | 5.89 | 4.71 | -20.0 | 0.66 | 3.76 |
| | $PM_{10}$ | 13.02 | 8.90 | -30.9 | 0.49 | 7.48 |
| Utö | | | | | | |
| | $NO_2$ | 3.25 | 2.32 | -28.7 | 0.51 | 1.95 |
| | $SO_2$ | 0.58 | 0.26 | -54.5 | 0.48 | 0.49 |
| | $PM_{2.5}$ | 3.93 | 3.23 | -18.0 | 0.54 | 3.02 |

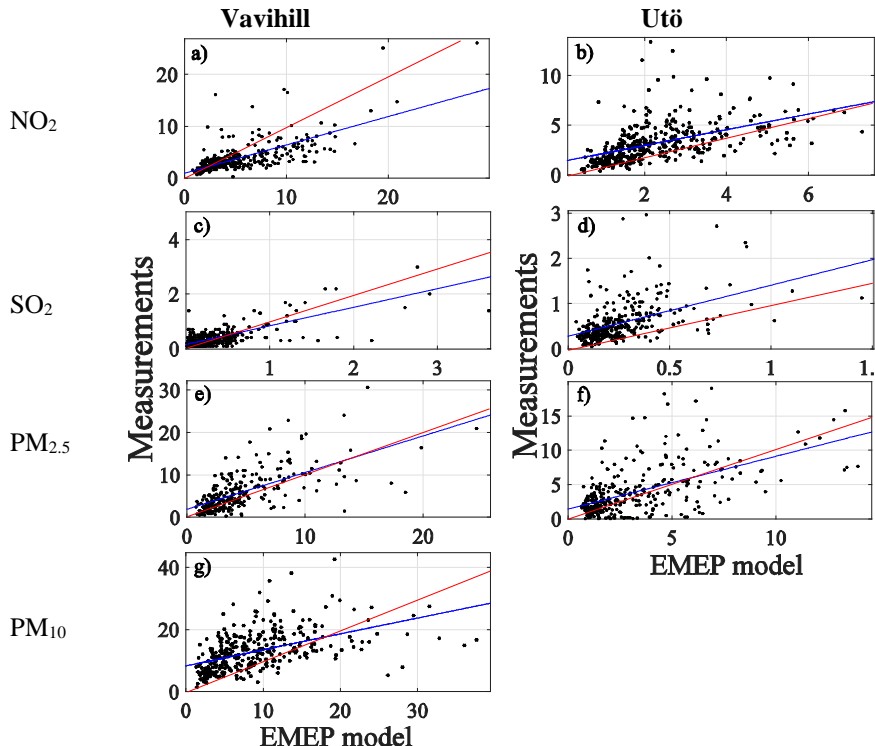

**Figure A1.** Scatter plots of model results versus measured data of daily average concentrations of $SO_2$, $NO_2$ and
particular matter at Vavihill (left) and Utö (right) in year 2013 ($\mu gm^{-3}$). The red line corresponds to a 1:1 ratio,
and the blue line shows the linear relationship between measured and modelled concentrations.

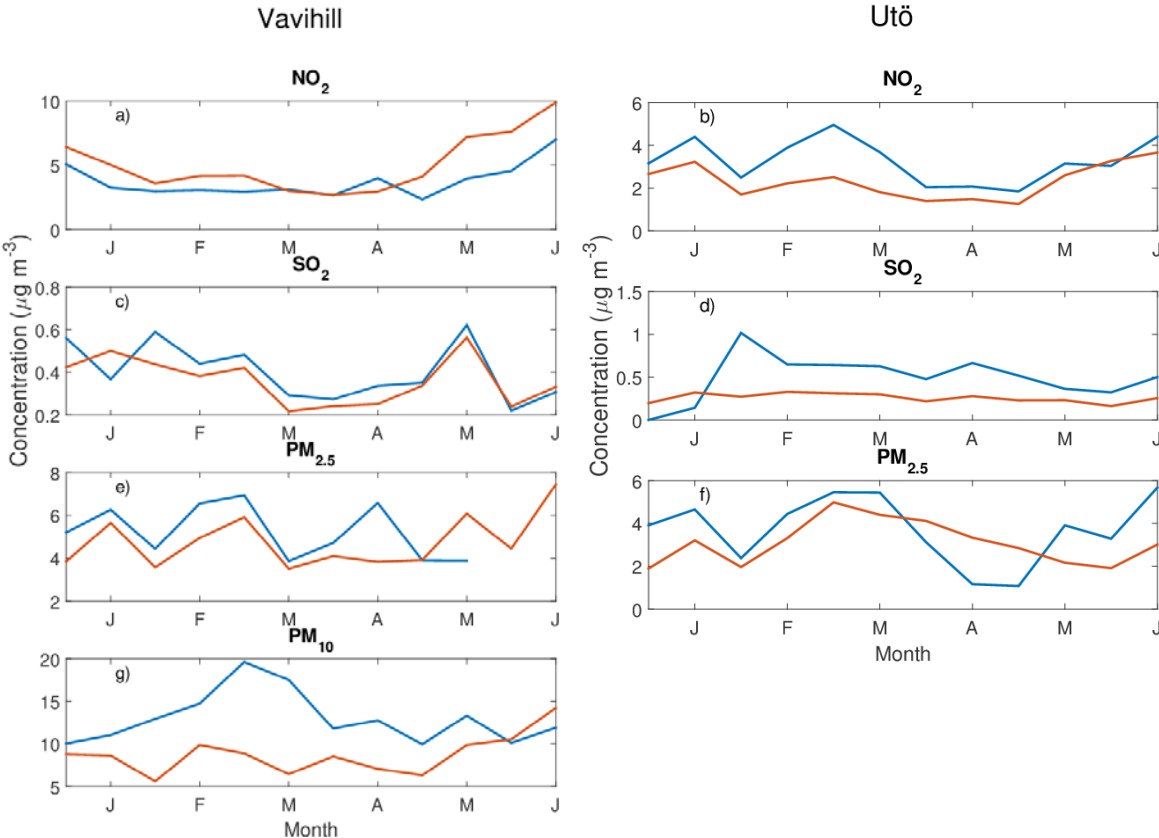

Vavihill     Utö

2 **Figure A2.** Measured and modelled monthly average of concentrations of the pollutants at Vavihill and Utö in

3 2013. The red line corresponded to concentrations of the EMEP modelling and the blue line showed measured

4 concentrations.

