# Peer review of "Contribution from Ship Emissions to the concentration and deposition of air pollutants in the Baltic Sea"

_Earth System Dynamics, 2016_

## Referee Comment (RC1) · Anonymous Referee #1 · 4 Jan 2017

This manuscript considers ship emissions and deposition in the Baltic and North Seas, using a chemical atmospheric transport model that forms part of the European Monitoring and Evaluation Programme (EMEP). The authors assess the temporal and spatial variations in emissions using EMEP and present the results.

I find the potential of these results interesting even though the current manuscript is rather limited - there is no data that takes into account the use of scrubber technologies, changing fuel types, new 2015 sulfur regulation, etc. However, much of the writing appears to be rushed and has not been read through by an experienced English speaker before submission. The language and clarity of much of the text needs to be significantly improved. In addition, the text is very light in detail and limited in discussion, which affects the readability and clarity of the paper. Certain points would benefit from more than one sentence to be absolutely clear about the message that the authors hope to convey.

The reader is often referred to other reports, databases and publications for much of the methods section. This is ok within reason, but I think it is too much in this case. For example, what is the methodology used in Omstedt et al 2015 (Section 3.3, page 7/8)? The method seems relevant enough to this paper that I think at least a summary of the detail should be included here.

Before this paper is accepted for publication, the readability of the text must be improved and more detail should be included throughout the manuscript.

Other comments: Page 1, Line 23: suggest 'type of scrubber' rather than 'amount of scrubber'

Page 2, Line 13-14: Sentence beginning 'The main reasons for these emissions....' does not make sense to me.

Page 2, Line 26-28: Other studies have come to different conclusions, in part due to the buffering capacity of seawater (e.g. Hunter et al 2011). This is worthy of more discussion because the Baltic has a reduced buffering capacity (as discussed later).

Page 6, Line 11: use of the word 'conformed' does not make sense

Page 6, Lines 10-16: it was not immediately clear that this paragraph is comparing the seasonal variability. I think more could be made of this – for example Uto data compares pretty well for much of the variables, especially the SO2.

Page 8, Line 14-15: Do you mean that the relative seasonal variability was retained? This sentence is unclear.

Page 8, Lines 25-28: The text here feels insufficient. How much lower are the SOx emissions? Where are they lower – in the N. Sea and Baltic in general, or in specific

regions? Also, I view SECA as a region, rather than an emission control level. In other words, the S emissions in SECA are different now compared to 2013. This should be stated very clearly, particularly as this paper appears not to model the 2015 regulations. . ..

Page 8, Lines 29-31: State the WHO guidelines first (and maybe in the legend of Fig. 6 as well).

Page 9, Lines 1-9: I found a lot of this text confusing, poorly worded and difficult to follow. There is no figure to look at for much of the text, so it is imperative that the text is clear.

Page 9, Line 9: Suggest new paragraph before 'In some areas..'

Page 9, Lines 15-16: No quantitative evidence is presented of the 'small variation'. Surely this could be done (either in absolute or percent terms)?

Page 10, Line 6: Again, no quantitative evidence is presented of the 'small variation'. Surely this could be done (either in absolute or percent terms)?

Page 10, Line 12: Why does the NOx show a seasonal cycle, but the SOx does not? This should be discussed.

Page 10, Line 24: This is unclear to me. If the modelling study INCLUDED maritime emissions, why would the actual emissions be higher?

Table 1: A lot of acronyms are used, which are confusing to the non-expert reader. Where possible these should be defined or (if projects or databases in their own right) weblinks should be provided.

Table 2: Not enough information here – what does tau represent?

Figures 2 and 3 could be substantially improved with additional labelling. Having Uto and Vavihill in the top row and NO2, SO2, PM2.5 and PM10 as the first column would help the reader immediately understand what was being plotted.

[Figure]

Figure 6: I recommend a clear delineation of the iso-line for the WHO PM10 and PM2.5 thresholds. This will clearly identify the regions that are exceeding the limits.

Figure 7: Label each plot with NOx, SOx, etc., rather than just in the figure legend.

Figure 9: Label each plot with Dry deposition of NOX, Dry deposition of SOx, etc., rather than just in the figure legend.

---

## Referee Comment (RC2) · Anonymous Referee #2 · 8 Jan 2017

Haglund et al., Earth Syst. Dynam. Discuss., doi:10.5194/esd-2016-46, 2016.

Temporal and Spatial variation of Contribution from Ship Emissions to the concentration and deposition of air pollutants in the Baltic Sea

Anonymous Review

General

The paper presents and discusses results of a numerical modelling study of the influence of international shipping on concentrations and deposition of pollutants (i.e. sulphur) in the Baltic and North Sea regions. The model used was the EMEP chemistry transport model with a grid resolution of about 50 km x 50 km, target years are

2009 to 2013. In addition, scenarios based on estimates of future conditions (different fuel sulphur content due to upcoming regulations) were also created and evaluated. Additionally the effect of open loop scrubber water release on the acidification of the Baltic Sea has been assessed. Unfortunately, the influence of a nitrogen emission control area (NECA), which has now been decided to come into effect in 2021, has not been considered. In general this is an interesting study addressing a relevant and current topic discussed by pollution assessments and in connection with international governance. But the paper seems not be ready for publication in the present form for several reasons. It could be published after some revisions.

Major remarks

i) It looks like that the text in parts has more the status of a draft paper (e.g. some untidy parts in the reference list and missing words), and it seems not to have been carefully read by someone, who is familiar with the English language (best a native speaker) before submission. There are many language issues which need to be clarified/corrected by someone experienced (the reviewer does not intervene here, since it is assumed that many text passages read differently after the paper has been revised). Often the line of thought is not easy to follow, especially in the sections describing methodological aspects.

ii) The paper cites the study of Jonson et al. (2015), which uses the same CTM (EMEP model, but with a higher spatial resolution) to evaluate the influence of shipping emissions on the air quality in the same region (Baltic and North Seas), discussing also future settings. In addition the current day shipping emissions in that paper are based on AIS data in combination with data from a technical ship data base, this is more or less state of the art. Here the present paper should tell the reader "what are the major differences between the two modelling exercises", i.e. to underline the added value of the present modelling study. It would helpful to learn why the relative coarse grid resolution has been used, although present day computing power should allow for better. It would also be interesting to learn why the authors decided for a non-AIS based ship

emission approach. It would very interesting to compare results of the two modelling studies (Jonson et al. and present), since they have overlapping modelled years and consider the same region. It should at least be tried to summarize differences and similarities in the results in more thorough manner.

iii) In the introduction to section 3 it is mentioned that the considered time period (2009 – 2013) was covered by two consecutive runs with different model versions (rv4.4 for 2009 to 2011 and rv4.8 for 2011 to 2013). Why this? It might be problematic. It is necessary to inform the reader on the differences between those model versions. Do they come to similar results, if the otherwise same settings are used? 2011 seems to have been modelled using both versions, this would an ideal case to compare the results. Please comment. Why has the period not been cover by a run using the newest version of the EMEP model?

Some minor remarks

Title: The title of the paper should include a hint that also future scenarios have been generated and evaluated. Why is the North Sea not in the title, although it has been modeled and is partly discussed by the paper? Is it modelled as source for advected pollutants with south westerly winds? At least in the beginning of the Abstract both seas are mentioned. If you have a special reason, why your evaluation concentrates on the Baltic Sea (although the NS is also modelled), please provide it in the introduction to the paper.

Abstract: In the Abstract a lot of numerical numbers are reported, which in general is good. Many of them are given as "maxium %", which is not so easy to conceive. To better grasp the pollution situation it is important for the reader to know: "Was this maximum (ore close values) reached several times in the period?" or was it a unique value (mean over all years?) in the distribution of concentrations and deposition values? Was it the result for a tiny spatial sector (it is anticipated that the smallest possible is 50 x 50 km2) or was a larger area affected? In short: Are the reported

maximum values relevant or are the extreme outliers? It would be better to provide also another statistical value (mean, std . . ., and max.) for high concentration areas and background. (To concentrate on reporting maximum values is also unfortunate, since on page 6, line 8, it is stated that the model has difficulties to represent maximum values. This was for Vavihill and Utö, but might be the case everywhere).

Nomenclature: The model used is the EMEP Chemistry Transport Model, it should be consequently named "EMEP model" throughout the text (and e.g. not just "EMEP", which is a programme).

Wrong section: On page 4, line 5 to 9; the information given here does not belong into a section called "Regulation of Shipping". Shift it.

Scope of paper: On page 2, line 17, it is written: "Here emissions of sulphurdioxide, nitrogen oxides and particulate matter are examined." No, this is not the case. The resulting concentrations and deposition values are examined. May be the effect of emissions are examined. This is not a work which concentrates on gaining better emissions.

Deaths: "Corbett et al. (2007) showed that shipping-related emissions of particulate matter contribute to approximately 60,000 deaths annually on a global scale,. . . ". Corbett et al. did not show this! It is at best a crude estimate.

New regulations: At the end of 2016 it has been internationally decided, that in 2021 a NECA will be implemented in the Baltic Sea and North Sea. In 2020 the Sulphur content in shipping fuels will be globally limited to 0.5%. This needs to be mentioned now in a future section 1.2. The latter value is relevant for your scrubber scenario reported at the end of section 3.2, where you still work with 2.7% S. Since the ships need to hold 0.5% globally, the owners will most prob. use fuel with this value (which is discussed for a longer time and now decided). Please comment.

Methodology: Please provide in section 3 a reason, why you choose the time period

2009 to 2013. Are there meteorological (e.g. variability) or computing limitation or emission data reasons?

2013: Please provide a rational, why a huge part of your discussion is addressing 2013. There might be a good reason (e.g. representative). Please discuss this selection more expanded.

Consistent: Last sentence of the paper in the summary: "The validation of the model showed that the model underestimated most of the pollutants but the model was overall consistent with the measured data in 2013 at Vavihill and Utö." How can this be (the model often underestimates)? Are these locations totally different compared to the domain. What does "consistent" mean here? Looking into figure 3 there may be some consistencies but as many inconsistencies. Please explain better your understanding of "consistency".

Figures: All figures should have units at their axis (only some have).

References:

page 2, line 13: the third reference should read "Matthias et al. 2010", also in reference list page 16, line 6 (Volker is the given name).

page 3, line 2: the references Arya (1999) and Raven and Berg (2006) are text books, an interested reader on the nitrogen impacts has to dig a lot in those books to find the relevant pages. Please provide more dedicated references here; there are many (and better ones).

page 14, line 4 to 9: here a few references (3) seem to be lumped together into one paragraph.

page 14, line 20: the paper by Jonson et al. 2015 is available as full ACP paper for 2015 (Vol. 15 /83-798), not ACPD.

page 16, line 10: it should be H. (Hulda) Winnes.

---

## Author Comment (AC1) · 8 Feb 2017

Dear editor

We have read the comments from the reviewers and agree with most of the comments and suggestions for improving the manuscript. We will rewrite and re-arrange sections to better keep the line of thought. The method to produce the deposition scenarios will be described more clearly. Also, the main focus of the manuscript will be transferred to the scenario study and the measurement study will be treated as validation and contribution to the discussion. The scrubber output will be more thoroughly discussed. Further, we will clarify and discuss the present study in relation to the Jonson (2015) paper, a very parallel and complementary (and party overlapping) study.

[Figure]

To reviewer 1 For clarification and emphasis reasons we will rewrite and re-structure parts of the MS. More discussions will be inserted and a more detailed description of the method used. The summaries or outcome from reports will be expanded. The language will be checked as well. Figures will be improved according to suggestions. Additional improvements: methodology section will be expanded/improved. Specific suggested changes and clarifications will be introduced.

To reviewer 2 We will work to better keep the line of thought and hence re-structure and rewrite some parts. The methodology will be clearer. Further the Jonson (2015) paper will be more thoroughly discussed in relation to our work. Differences in EMEP model versions will be discussed. For clarification and emphasis reasons we will rewrite and re-structure parts of the MS. More discussions will be inserted and a more detailed description of the method used. The summaries or outcome from reports will be expanded. The language will be checked as well. Figures will be improved as suggested. We agree that the title could be changed, our suggested new title is: "Present and future variation of Contribution from Ship Emissions to the concentration and deposition of air pollutants in the Baltic Sea" We agree that the North Sea is partly included, but not in the scenarios, this is why we would like only the Baltic Sea to be in the title. We will have a more through explanation on the choices of years and the model versions used. For the minor comments we agree with the reviewer's suggestions and will change accordingly.

---

## Author Response (AR1)

**Revisions in manuscript:**

**Present and future variation of Contribution from Ship Emissions to the concentration and deposition of air pollutants in the Baltic Sea**

**Authors**

We have changed the order of the authors as

 Björn Claremar, Karin Haglund, Anna Rutgersson

**Abstract**

This part is completely rewritten and changed by emphasizing the scenario parts and to present an overview of the spatial pattern and comparison to measurements. Most specific numbers are condensed.

Specific comments on reviewers' remarks

- Most specific numbers are condensed because of the uncertainties in extreme values

**1. Introduction**

This part is more or less completely rewritten. First paragraph is left slightly changed while the rest is partly re-arranged for the line of thought and partly renewed to shift the focus towards the scenarios. We also present more thoroughly the study of Jonson et al. (2015) and point out the novelty of the present study and that we will compare our results to theirs regarding effect of model resolution and nitrogen scenarios.

Specific comments on reviewers' remarks

- The Scope is clarified together with the novelty
- Jonson et al. (2015) study is presented in more detail and so is our relation to that work, with comparisons regarding concentration and deposition patterns
- NECA is discussed and is implemented in the scenarios and analysis
- Also the 0.5% S in global EU fleet is discussed

**2. Data**

In this section clarifications and rearrangement are made. The names of the subsections are now EMEP model system, EMEP model data, Other data sources and Measurements.

Specific comments on reviewers' remarks

- Data and report content description is expanded.
- Model versions rv.4.4 and rv.4.8 is compared

**2.1 EMEP model system**

Model performance evaluation by Gauss is moved here from (old section 2.3).

**2.2 EMEP model data**

Text is somewhat expanded to clarify. Also the differences between model versions rv.4.4 and 4.8 are shown (also with figure).

- Added new figure:
    - Difference (in %) of deposition from EMEP model rv.4.4 to rv.4.8 for a) oxidized sulphur and b) oxidized nitrogen.

**2.3 Other data sources**

This is a new section that presents data from emission databases and results from the MATCH model, used for the background deposition scenarios in the future.

**2.4 Measurements**

The old "model performance" section is renamed to "measurements" to present only the Vavihill and Utö concentration data.

**3. Methods**

This is the renamed methodology section. We changed some wording of the section headers and added a section "Model performance of concentrations".

Specific comments on reviewers' remarks

- The reason for the use of different model version is presented
- The method is presented more in detail
- Line of thought is improved

**3.1 EMEP Model Runs**

This part is rewritten to clarify why the two versions of the EMEP model is used.

**3.2 Model performance of concentrations**

This part consists of parts from old section 2.3 (Model Performance, related to the comparison of simulated and observed concentrations.

**3.3 Future Ship Emissions**

Apart from some minor wording changes, the last paragraph about sulphur reduction is removed. We also inserted a sentence about NECA.

**3.4 Deposition scenarios of ship emissions**

Here we made the text clearer.

**4 Results**

We have restructured this section more or less completely. Some figures and tables are added.

**4.1 Ship deposition scenarios**

Text is very much expanded because of a deeper analysis. First we present the distribution of ship emission and associated deposition in the base year. Also the associated seasonal variation of the ship deposition in different Baltic Sea basins are presented.

Then the emission and scrubber scenarios are presented and discussed, to be followed by the estimated deposition scenarios from ship traffic and also related to other sources. Totals for the whole Baltic Sea area is presented as well in a table. The outline for the deposition is the separation between oxidised sulphur and nitrogen, with the former presented first. For the nitrogen first the original data is presented and then the influence from NECA is implemented. The proton input (2 for S and 1 for N in nmol $m^{-2}$ $yr^{-1}$) in the Baltic Sea basins is estimated for the different scenarios. Last a correction factor is also calculated to convert rv.4.4 results to rv.4.8 results. However, the impact is estimated to be small, at the larger scale, and not used.

Specific comments on reviewers' remarks

- New calculations with NECA
- 0.5 % S in fuel globally is discussed

Scenario figures with monthly resolution are replaced by annual resolution with relative monthly changes in a separate figure.
New figures:
- Total emissions of $SO_2$ and deposition of OXS from international shipping in the Baltic Sea and North Sea in 2011.
    - Partly new
    - replaces partly old fig. 5 and fig. 8
- Monthly deposition of oxidized sulphur (OXS) in six basins.

    - New

- Annual ship deposition of sulphur ($mgm^{-2}$) in six basins
    - Replaces the monthly data in old fig. 10

- Annual deposition of sulphur from all emission sources ($gm^{-2}$) in six basins of the Baltic Sea year 1900 to 2050.
    - Replaces the monthly data in old fig. 11
- Annual ship deposition of nitrogen ($gm^{-2}$) in the basins of the Baltic Sea, year 1900 to 2050.
    - Replaces the monthly data in old fig. 12
- Annual deposition of nitrogen from all emission sources ($gm^{-2}$) in the basins of the Baltic Sea year 1900 to 2050.
    - Replaces the monthly data in old fig. 13
- In a) estimated part of fleet applying to TIER I (blue), TIER II (red) and TIER III/NECA (orange), in b) correction factor for OXN deposition from shipping, using the implementation of TIER II and the TIER III in NECA from 2021.
    - new
- Annual proton input from OXS and OXN in the basins of the Baltic Sea (defined in Fig. 2) for year 2010 to 2050. The red line corresponds to Shipping scenario 3, the magenta and cyan line to Shipping scenario 4 and 5, respectively (scrubber + atmospheric deposition). The black line shows historical shipping (derived in Omstedt et al., 2015). Green and blue lines is with implantation of TIER II and NECA and scrubber scenario 4 and 5, respectively.
    - new
- Correction of deposition caused by ship traffic, in the different Baltic Sea basins.
    - new

New tables:

- Total deposition of OXS and OXN in Baltic Sea Mg yr$^{-1}$.
    - New
    - Partly replaces fig 14-15

- Emissions from shipping in the Baltic Sea in Mg yr$^{-1}$.

    - new

**4.2 Present emissions and surface concentrations**

This section replaces old section 4.1 (Surface concentration) and adds discussion of present emissions.

The concentration validation table and figures are moved to an appendix.

New figures

- Annual mean concentration of near-surface concentration (at 3 m level) of particulate matter from all emission sources in the EMEP area, in upper panel for rv.4.4 and in lower panel in rv.4.8.
    - new
- Percentage (%) of the total surface concentration, caused by international shipping in the Baltic Sea and the North Sea in 2011
    - New colorbar
- Concentration of PM$_{2.5}$ caused by shipping
    - New colorbar

**4.3 Present Deposition**
This section replaces old section 4.2 (Deposition). It analyses the simulated deposition in 2013. Text is re-arranged to more clearly follow the line of thought. Some comparison is also made with Johnson et al. (2015).

- Deposition of a) OXS and b) OXN caused by shipping.
    - Replaces old fig. 8
- Percentage (%) of the deposition, caused by international shipping in the Baltic Sea and the North Sea in 2013 of- (a) Dry OXN, (b) Dry OXS, (c) Wet OXN, (d) wet OXS.
    - Replaces old fig. 9

**5 Discussion**
This section is expanded and re-arranged to get a clearer line of thought. First we discuss emissions, regulations and the potential effect of using scrubbers on acidification. Then we turn to uncertainties with ship emissions, the methods used regarding resolution of the model and statistical approach. Last some aspects interesting for future studies are discussed.

**6 Summary and Conclusions**
This section is expanded regarding the scrubbers and scenarios. The section ends with conclusions in bullet form.

**References**

Specific comments on reviewers' remarks

- The text books Arya and Raven & Berg are completely removed.
- The other errors are corrected

**Figures and tables**

Specific comments on reviewers' remarks

- Tau correlation is removed from the (old) table 2.
- The figure colour bars are made simpler to read.
- Units and headers are updated
- The PM10 is not trespassed at all, and PM2.5 is trespassed only in the Benelux area, also now mentioned in the text, but not shown in the figure.

**Appendix: Concentration validation at Vavihill and Utö**

---

## Author Response (AR2)

**Letter to editor**

- Reference "J15" is unusual; it is up to the publishers to allow this abbreviation.
    - We keep it and wait for the publishers opinion
- p3, line 26: Ref. should be "Jonson" not "Jonsson", or vice versa in ref list, whichever is correct
    - OK
- Figures 1, 13-17: land contours are invisible, should be improved
    - We have made the coastlines thicker in the above figures and also in Fig. 3.
- All figures showing the same scenarios: if possible, same colours should be used for the same scenarios
    - The issue covers Figs. 5-9 and 11. Fig. 5 is now changed to correspond to the colours given in the other figures
- p25, line 34, delete "is"
    - OK
- Consider if you may want to change the title to better describe your key message; if I understood correctly, that scrubbers may increase Baltic Sea acidification. The title as is now is rather general and sounds not very interesting. A suggestion could be "Ship emissions and the use of current air cleaning technology: Contributions to air pollution and acificication in the Baltic Sea". Or similar. This is just a suggestion, so feel free to use the title you feel is appropriate for your paper.
    - We followed your advice.

Further the text was re-read, typos were corrected and some rephrasing was done.

**Letter to reviewer 1**

- Section 'Introduction', page 10, line 13ff:
  Work done by J15 and differences to the present study are now better explained, but a clear statement, which justifies the present work is still missing. It might be referred to the benefit of many more meteorological years analysed here.
  - o We add: ". The use and averaging of 3 years (2009–2011) for the present deposition fields reduces the variability from meteorology for the future scenarios."

- "Section 2.2 'EMEP Model Data', line 29 ff:
  It is stated that several modifications have been made concerning aerosols. State, why you have made these modifications. Sounds substantial. Is it now not the standard EMEP model any more? Provide references for e.g. new parameterisations and justify newly selected rates.
  - o We did not do the changes ourselves. We clarify by: "Between model versions several changes that affect aerosol production/modelling have been implemented by the EMEP community; e.g. modification of the sea salt parametrization…"

- Line 21: is there a reference for harbour activity questionnaires?
  - o We add the reference (ENTEC, 2002) in the sentence.

- Section 3.4 'Deposition scenarios of ship emissions' is still not easy to read, it appears a bit confusing. Try to make points clearer and justify assumptions (like e.g. the '50%' in line 24 on page 10).
  - o We have revised the text in the section.
  - o Regarding the 50% we write: "where 50%, as a first-order approximation, was assumed to depend on emission trends in the North Sea".

- Section 4.2, page 19, lines 5 and 6: it reads like year 2011 is compared to year 2011 and judged to be the same (??); what 'distribution' is meant?
  - o We write: "The emission distribution (relative) is almost the same in 2013.

- "Figures:
  Figures 13 to 17 need to be redrawn, in many of the Baltic Sea coastline is hidden behind intense foreground colours, and therefore the geographical orientation is not easy. (technical issue)
  - o We have made the coastlines thicker in the mentioned figures and also in 1 and 3.

- In figure 6 sub-plot for BB: scenarios 4 and 5 are missing. Why?
  - o We add in the figure text: "It is here assumed that scrubbers are not used in Bothnian Bay."
  - o At p. 12, line 10-11 we clarify with: "Note that it is assumed that ships are not using scrubbers north of Baltic proper (as discussed in section 3.3).". See p. 9, line 25-26

- Language:
  Language issues have been improved, but there still are many omitted definite articles, plural and missing little words (too numerous to be listed here; i.e. sec. 4.2). Here the paper needs careful language editing.
  - o The text was re-read, typos were corrected and some rephrasing was done.

**Letter to reviewer 2**

- A general note on figures - please make all coastlines a thicker and/or different colour. I found that the combination of coastlines and contour lines all in the same line style very disorientating when looking at the plots.
  - We have made the coastlines thicker in the figures 1,3 and 13-17.

- Page 2, Line 13 Should be sulphur 'dioxide'
  - OK

- Page 2, Line 27 should be 'both act as acidifying compounds'
  - Ok

- Page 4, Line 2 typo - I think 'hand' should be 'and'
  - OK

- Page 4, Line 9 This sentence does not make sense:'Concentrations and depositions were analysed valid for 2010 and show 9 concentrations along the shipping routes for concentrations but a significant spread for depositions.'
  - Changed to: "The concentrations and depositions were analysed for the year 2010. Concentrations were focused along the shipping routes but there was a significant spread for depositions."

- Page 4, Line 19 Comma required after 'cleaning technologies'
  - OK

- Page 5, Line 2 Remove the word 'the' from between 'In' and 'Gauss et al., 2015'
  - OK

- Page 5, Line 5 Replace 'underestimates' with 'underestimated'
  - OK

- Page 13, Line 12 Suggest changing the text to say: 'the Baltic, but the levels will stay low.'
  - OK

- Page 14, Line 5 remove word 'is'
  - OK

- Page 15, Line 17 Replace 'implantation' with implementation'
  - OK

- Page 16, Line 3 remove word 'is'
  - OK

- Page 16, Line 11/12 TIER III 'will be introduced in 2021.'
  - OK

- Page 17, Line 13/14 I assume that 4x10^-6 and 7x10^-6 refer to pH changes? This needs to be clarified as it is not clear at the moment.

- o Changed to: "At pH 8, a proton input of 1 nmol $m^{-2}$ corresponds to a decrease in pH of $4\cdot10^{-6}$ for a mixed ocean surface layer of 10 m."

- Page 20, Line 1 Replace 'underestimating' with 'underestimation'
  - o OK

- Page 20, Line 4 Change to 'guidelines for the annual average exposure for PM2.5 = 10 ug m-3, PM10 = 20 ug m-3...' (etc)
  - o OK

- Page 21, Line 8 Change to 'was a maximum of around 20% and, for PM10, 13%.'
  - o Changed to: "a maximum of around 20% and, for $PM_{10}$, 13%."

- Page 21, Line 9 Remove 'as'
  - o OK

- Page 22, Line 20 Change to 'are in-line'
  - o OK

- Page 24, Line 10 Change to 'Similar deposition scenarios were used in Turner et al. (2017),'
  - o OK

- Page 24, Lines 13/14 This needs rephrasing. I think the authors are saying that the impact of NECA is greater when scrubbers are not in use? i.e. that scrubbers offset the benefits of NECA?
  - o We instead write: "The conclusion that scrubbers increase the ocean acidification still holds, but it is decreased by less than 20 %, when including the effect from NECA. Without scrubber, the impact from NECA is very large on reduced acidification In other words, scrubbers offset the benefits of NECA."

- Page 24, Line 18 Replace 'but' with 'which is'
  - o We rephrase: "Locally, along shipping lanes, the pH decrease can be comparable to $CO_2$ uptake (Fig. 12a)."

- Page 25, Line 22 Change to 'Stricter regulations of sulphur content in maritime fuel and increased use of other fuels will result in...'
  - o OK

- Page 25, Line 36 Change to 'was a maximum of around 20% and, for PM10, 13%.'

[revised manuscript text omitted]

**Kommenterad [AR1]:** Det är fortfarande väldigt manga figurer. Frågan är om man inte skulle lägga jämförelsen med mätdata (+tabell och 2 figurer) i ett appendix?

| Station | Component | Obs. ($\mu$gm$^{-3}$) | Mod. ($\mu$gm$^{-3}$) | Bias (%) | Corr. (r) | RMSE |
|---------|-----------|------|------|------|------|------|
| Vavihill | | | | | | |
| | NO$_2$ | 3.69 | 5.05 | 36.7 | 0.72 | 3.03 |
| | SO$_2$ | 0.42 | 0.38 | -8.2 | 0.70 | 0.35 |
| | PM$_{2,5}$ | 5.89 | 4.71 | -20.0 | 0.66 | 3.76 |
| | PM$_{10}$ | 13.02 | 8.90 | -30.9 | 0.49 | 7.48 |
| Utö | | | | | | |
| | NO$_2$ | 3.25 | 2.32 | -28.7 | 0.51 | 1.95 |
| | SO$_2$ | 0.58 | 0.26 | -54.5 | 0.48 | 0.49 |
| | PM$_{2,5}$ | 3.93 | 3.23 | -18.0 | 0.54 | 3.02 |

[Figure]

**Figure A1.** Scatter plots of model results versus measured data of daily average concentrations of SO$_2$, NO$_2$ and
particular matter at Vavihill (left) and Utö (right) in year 2013 ($\mu$gm$^{-3}$). The red line corresponds to a 1:1 ratio,
and the blue line shows the linear relationship between measured and modelled concentrations.

[Figure]

**Figure A2.** Measured and modelled monthly average of concentrations of the pollutants at Vavihill and Utö in
2013. The red line corresponded to concentrations of the EMEP modelling and the blue line showed measured
concentrations.